# Stable memory with unstable synapses

Lee Susman[1,2]*, Naama Brenner[2,3]* & Omri Barak [2,4]*

What is the physiological basis of long-term memory? The prevailing view in Neuroscience attributes changes in synaptic efficacy to memory acquisition, implying that stable memories correspond to stable connectivity patterns. However, an increasing body of experimental evidence points to significant, activity-independent fluctuations in synaptic strengths. How memories can survive these fluctuations and the accompanying stabilizing homeostatic mechanisms is a fundamental open question. Here we explore the possibility of memory storage within a global component of network connectivity, while individual connections fluctuate. We find that homeostatic stabilization of fluctuations differentially affects different aspects of network connectivity. Specifically, memories stored as time-varying attractors of neural dynamics are more resilient to erosion than fixed-points. Such dynamic attractors can be learned by biologically plausible learning-rules and support associative retrieval. Our results suggest a link between the properties of learning-rules and those of network-level memory representations, and point at experimentally measurable signatures.

[1] Interdisciplinary Program in Applied Mathematics, Technion Israel Institute of Technology, Haifa 32000, Israel. [2] Network Biology Research Laboratories, Technion Israel Institute of Technology, Haifa 32000, Israel. [3] Dept. of Chemical Engineering, Technion Israel Institute of Technology, Haifa 32000, Israel. [4] Rappaport Faculty of Medicine, Technion Israel Institute of Technology, Haifa 32000, Israel. *email: lee.susman@gmail.com; nbrenner@technion.ac.il; omri.barak@gmail.com

The ability to form and retain memories of past experience is fundamental to behavior, supporting adaptable responses, and future planning[1]. These internal representations persist over extended durations and may be reactivated by appropriate retrieval cues[2]. Currently, it is widely accepted that synaptic connections between neurons play a central role in the physiological basis of long-term memory storage[3] (see refs. [4,5] for other possibilities). The process of learning, on its part, is understood as stimulus-driven neural activity sculpting network architecture, i.e., Hebbian plasticity[6].

If an internal memory-representation is stable over time, then one could assume that some properties of its underlying neural implementations also exhibit invariance over this period. However, at the level of single synapses, no such robustness exists (reviewed in refs. [7–10]). Over the past decade, several studies, both ex vivo[11,12] and in vivo[13], suggested that synapses undergo significant spontaneous changes. These fluctuations persist even in the absence of neural activity, with magnitude estimated to be as large as that of directed, Hebbian, plasticity[14].

How, then, can memory traces remain stable over time? What remains invariant while all synaptic strengths vary? Various studies have proposed candidate invariant features, at different levels of organization of neural networks. For single synapses, invariance may be implemented in a sub-set of the largest spines[13,15]. Invariance may, instead, only emerge at the level of the connection between neurons, typically comprising several synapses. This allows individual synapses to fluctuate, under the constraint of stable overall connection strength between two cells[16,17]. Higher up the organizational hierarchy, invariant features may manifest only at the network level. This would allow individual connections to fluctuate, while at the same time some network properties remain stable[9].

In this work, we suggest that such network-level invariance arises naturally when considering the effect of homeostatic mechanisms on network stability. Stability is best understood by considering a global property of the network connectivity matrix—its eigenvalues. This is a collection of points in the complex plane, each having a real and imaginary part. We show that under activity-independent synaptic fluctuations and known homeostatic mechanisms, information encoded in the real part of these eigenvalues is strongly eroded, while imaginary-coded information is spared. Consequently, in a system that utilizes imaginary-coded memory, single synapses may exhibit ongoing fluctuations whereas global features remain invariant as network-level properties.

We investigate this concept by showing how different homeostatic plasticity mechanisms degrade real- or imaginary-coded memories, and how spike timing dependent plasticity (STDP) can store transient inputs as imaginary-coded memories. We then show the implications of such memories—that learned representations give rise to stable oscillatory trajectories of network activity. These memory states can be viewed as the time-varying analogs of stable fixed points in the classic Hopfield model[18]. Having been learned and embedded in a component of connectivity, memory items may be transiently retrieved by supplying an associative recall cue.

Our results suggest a principle by which memory can be learned and retained in a stable manner despite significant ongoing synaptic fluctuations. The implications of such a mechanism to experimental data are discussed both in terms of measured neural activity and in terms of synaptic plasticity during learning as opposed to at rest.

## Results

**Model**. Our model is based on a standard framework of firing-rate neural networks[19]. It consists of $N$ recurrently connected neurons,

with $W_{ij}$ the synaptic connection strength from neuron $j$ to $i$. Each neuron $i$ transforms its input $x_i$ into firing rate via a nonlinearity $\phi(x_i)$, where the state vector $\mathbf{x} = (x_1\ x_2\ \cdots\ x_N)^\top$ evolves as

$$\dot{\mathbf{x}} = -\mathbf{x} + \mathbf{W}\phi(\mathbf{x}) + \mathbf{b}(t), \qquad (1)$$

and $\mathbf{b}$ is an external input. Here and below we denote by $\phi(\mathbf{x})$ the vector obtained by applying $\phi$ to each coordinate of $\mathbf{x}$.

Connectivity of task-performing networks is often designed to achieve a desired functionality, and assumed to be constant while the network is performing the task[18,20]. There are models in which connectivity co-evolves with neural dynamics, but changes are usually confined to a training phase, whereas connectivity is kept constant during the test phase[21,22]. These models are consistent with the expectation of synaptic tenacity in the absence of learning. In our model, to incorporate the recent observations on synaptic fluctuations, the connectivity matrix $\mathbf{W}$ continuously co-evolves with neural activity $\mathbf{x}$ throughout all task phases, albeit with a slower timescale (Fig. 1, see also ref. [23]).

In order to study the coexistence of memory with synaptic fluctuations, we let $\mathbf{W}$ evolve due to contributions arising from both learning-related and fluctuation-related terms, denoted by $\Delta_L$ and $\Delta_F$ respectively:

$$\dot{\mathbf{W}} = \eta(\Delta_L + \Delta_F), \qquad (2)$$

with $\eta > 0$ the plasticity rate (relative to neural dynamics). The fluctuation term includes stochastic, activity-independent noise in synaptic strength, as well as a homeostatic mechanism to control synaptic and firing-rate stability. These are precisely the processes which endanger the resilience of an acquired memory that is assumed to be stored in synaptic patterns. We first consider how an existing memory is eroded by these processes, and later address the learning part and the interplay between the two.

**Homeostatic plasticity erodes real-coded information**. We model spontaneous activity-independent synaptic fluctuations by a white noise process $\xi_{ij}$ driving each synapse $ij$ independently. The variance of such a process grows without bound, and thus, without a restraining mechanism these fluctuations would lead to divergence of the synaptic weights $W_{ij}$. A plausible restraining agent is homeostatic plasticity, modeled here as a network-level mechanism that stabilizes both synaptic weights and neural activities on average over long timescales. To understand this stabilization, we turn to dynamical systems theory that determines stability about a set-point by the spectrum of the appropriate Jacobian matrix (which is the local linear approximation of the dynamics). This spectrum consists of a set of complex

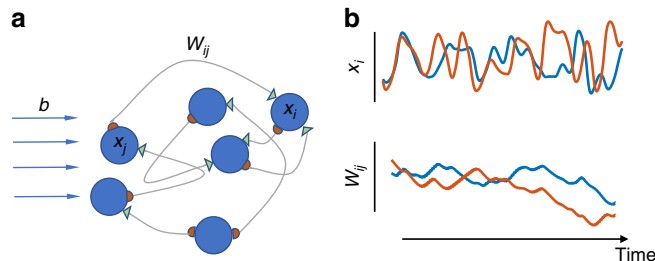

**Fig. 1** Co-evolution of neural activity and connectivity. **a** Illustration of our modeling framework: a recurrently connected neural network, with $W_{ij}$ denoting the connection strength from neuron $j$ to neuron $i$. The dynamic variables $x_i$ evolve by Eq. (1) and connection strengths $W_{ij}$ evolve by Eq. (2). An external signal $b_i$ can be added as input to each neuron $i$. **b** Two example traces of neural state dynamics ($x_i(t)$; top, colors) and of connection strength dynamics ($W_{ij}(t)$; bottom, colors) are shown. Both evolve over time, though on different timescales, and their dynamics are coupled

eigenvalues—a collection of points in the complex plane, each having a real part and an imaginary part.

In general, the real part of this spectrum defines the system's stability: a system is only stable if all its eigenvalues have negative real parts. The imaginary part of the spectrum, in contrast, determines the typical timescales of small-amplitude dynamics around this set-point, but not stability itself (Fig. 2a). Therefore, while the real part of the spectrum must be under the control of homeostatic plasticity, its imaginary part is not constrained by the requirement of stability, and is free to store information (Fig. 2b).

The arguments above derive from a general intuition on system stability; they are not a mathematical proof, as they depend on the existence of a set-point and its exact properties. They do, however, provide motivation to test this idea using various homeostatic mechanisms. We perform such tests using the connectivity matrix $\mathbf{W}$ as a proxy for the Jacobian. In the case of a linear network, or of linearizing around the origin, the two are equivalent. Our results below indicate that such an approximation is useful also in more general cases.

The embedding of a memory item is often represented in learning theory as a low-rank perturbation to the connectivity matrix, $\mathbf{W} \rightarrow \mathbf{W} + \delta\mathbf{W}$. For example, in the Hopfield model a memory is associated with a particular pattern of activity, a vector $\mathbf{u} \in \mathbb{R}^N$, and is embedded in connectivity by adding a perturbation of the form $\delta\mathbf{W} = \mathbf{u}\mathbf{u}^\top$. Such a perturbation is symmetric, $\delta\mathbf{W}_{ij} = \delta\mathbf{W}_{ji}$, and modifies the connectivity spectrum to include a real positive eigenvalue (with an eigenvector in the direction of $\mathbf{u}$). To tap into the resilience of the imaginary part of the spectrum as argued above, one might add an anti-symmetric perturbation ($\delta\mathbf{W}_{ij} = -\delta\mathbf{W}_{ji}$) that gives rise to an imaginary conjugate pair of eigenvalues. This defines a different type of

memory item, the simplest form being $\delta\mathbf{W} = \mathbf{u}\mathbf{v}^\top - \mathbf{v}\mathbf{u}^\top$. If the above general arguments on system stability are correct, such memory items should be more resilient to synaptic fluctuations. We test this by comparing the erosion of the two types of memory items under various homeostatic mechanisms. We first embed memories corresponding to either real or imaginary eigenvalues into the connectivity matrix $\mathbf{W}$, and then follow the dynamics of Eqs. (1) and (2) without active learning ($\Delta_L = 0$), but with various homeostatic models in $\Delta_F$.

Perhaps the simplest implementation of a homeostatic mechanism is by dissipative synaptic dynamics. Together with the noise $\xi$, this gives a fluctuation term

$$\Delta_F = \xi - \beta\mathbf{W}, \qquad (3)$$

with $\beta > 0$ the rate of dissipation. Figure 2c shows the eigenvalues of the connectivity matrix as a function of time (gray lines), with the eigenvalues corresponding to the memory highlighted in green. It is seen that the memory representation rapidly decays for both real (top) and imaginary (bottom) eigenvalues. This is expected from a dissipative system, where all information decays exponentially with a rate $\beta$. Therefore, in the presence of such a mechanism, neither type of memory items can be sustained for longer than the decay time $1/\beta$. However, as will be shown below, this is not the case for more indirect homeostasis mechanisms.

A biologically plausible homeostasis mechanism can be modeled as an activity-dependent rule—where the synaptic matrix is modified to achieve a stable post-synaptic firing-rate[24–26]:

$$\Delta_F = \xi + (\phi_0 - \phi(\mathbf{x}))\phi(\mathbf{x}^\top) \circ \mathbf{W}, \qquad (4)$$

with $\phi_0$ an arbitrary target-rate vector, and $\circ$ denoting a

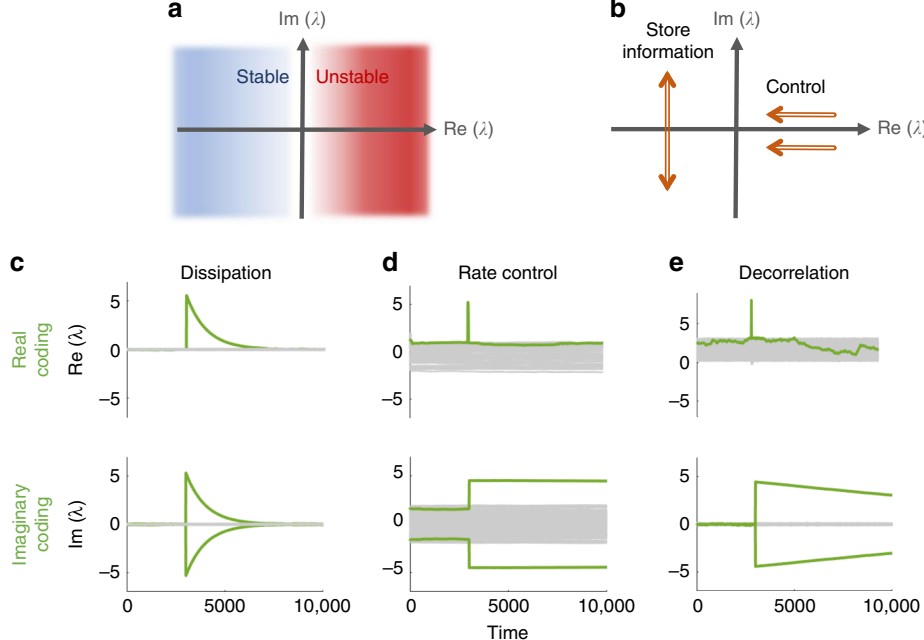

**Fig. 2** Stability and memory-associated connectivity eigenvalues. **a** Eigenvalues of the Jacobian matrix occupy the complex plane. System stability about a set-point is ensured if all eigenvalues have negative real parts (i.e., reside in the blue half-plane). Eigenvalues with positive real parts (in the red half-plane) correspond to locally unstable directions. **b** Homeostatic control that prevents noise from accumulating and causing divergence, also affects stored memories represented by the eigenvalues. However, such mechanisms must only control their positive real parts, pushing them to be negative (arrows), while in the left half-plane large-amplitude imaginary eigenvalues can persist. **c–e** Real (top) and imaginary (bottom) parts of connectivity ($\mathbf{W}$) eigenvalue spectra for systems evolving under the dynamics of Eqs. (1) and (2), with different homeostatic mechanisms in $\Delta_F$ ($N = 128$, see "Methods" for simulation details). For each case, a memory item was embedded in $\mathbf{W}$ at time $t = 2500$ (green trajectories), by inducing a real outlier eigenvalue (top panels, generated by the perturbation $\delta\mathbf{W} = \mathbf{u}\mathbf{u}^\top$) or an imaginary pair (bottom panels, generated by $\delta\mathbf{W} = \mathbf{u}\mathbf{v}^\top - \mathbf{v}\mathbf{u}^\top$). **c** Dissipation of synapses. Both real and imaginary memories decay with the same rate. **d** Homeostatic rate-control. **e** Decorrelation homeostasis. In the last two, real-coded memory (top) decays rapidly whereas imaginary-coded (bottom) persists

Hadamard (element-wise) product. Stabilization of firing rates around the set-point $\phi_0$ is achieved in this case by scaling the weight of each synapse by a factor dependent on its pre- and post-synaptic neurons. Although not explicitly symmetric (which would lead to a change in real eigenvalues), the stabilization induced by this rule requires control over the real part of the relevant Jacobian. Accordingly, Fig. 2d (top) shows that memories stored as real eigenvalues of $\mathbf{W}$ rapidly decay. Imaginary-coded memories, on the other hand, may persist indefinitely without interfering with homeostasis (Fig. 2d, bottom).

Finally, inspired by ref. [27], we consider a homeostasis mechanism that does not have a well-defined firing-rate set-point. Instead, this rule contains an anti-Hebbian term that reduces the connections between correlated neurons, thus pushing towards decorrelated firing rates across the network:

$$\Delta_F = \xi + \mathbf{I} - \phi_{\text{post}}(\mathbf{x})\phi_{\text{pre}}(\mathbf{x}^\top), \tag{5}$$

where $\phi_{\text{pre}}, \phi_{\text{post}}$ are two sigmoidal functions and $\mathbf{I}$ is the identity matrix. This rule leads to persistent and constrained fluctuations in both connectivity and firing rates[27]. The activity $\mathbf{x}$ is dominated by the unstable modes of $\mathbf{W}$, which are then suppressed by the anti-Hebbian term, leading to a new unstable mode in an endless succession.

Once again, we find that the decay of imaginary-coded memories is orders of magnitude slower than that of real-coded ones (Fig. 2e). Note that if the sigmoidal functions are identical, $\phi_{\text{pre}} = \phi_{\text{post}}$, this rule can only modify the symmetric part of $\mathbf{W}$. In practice, for many non-identical choices of the sigmoid functions, the modification is still mostly symmetric. Nevertheless, the relative decay of imaginary- and real- based memories is similar to the case of the rate-control rule, that does not have any symmetric tendency.

Some evidence for the generality and limits of validity of these results is presented in Supplementary Fig. 1, where sparse networks are considered. The amplitude of embedded imaginary memories decreases smoothly as the network becomes sparser; they remain more resilient than real-coded memories for the entire regime tested.

In light of these results, a natural question arises: can a dynamical learning rule utilize the imaginary subspace to robustly code and store memory representations? Perhaps surprisingly, we find that Spike Time Dependent Plasticity (STDP), a well

documented and biophysically plausible learning rule, provides a natural candidate for learning imaginary-coded memories.

**STDP stores imaginary-coded information.** Symmetric and anti-symmetric matrices give rise to real and imaginary eigenvalues respectively. It is thus reasonable that an anti-symmetric modification to the synaptic weight matrix $\mathbf{W}$ would primarily lead to changes in the imaginary part of its spectrum. Local learning rules observed in experiments (e.g., STDP) have a well-defined directionality: consecutive firing of neuron $j$ before $i$ leads to a strengthening of the connection $W_{ij}$ and to the weakening of the reverse connection. The temporal asymmetry of STDP[28] leads to an approximately anti-symmetric learning rule when applied to our rate model (see "Methods"); as such, this rule mostly affects the imaginary part of the spectrum. In the case of perfect anti-symmetry, we find the form $\Delta_L = \phi\mathbf{y}^\top - \mathbf{y}\phi^\top$, which modifies only the anti-symmetric component of $\mathbf{W}$. The vector $\mathbf{y}$ is a smoothed version of the activity $\phi(\mathbf{x})$, and arises in our rate-based formulation when applying an exponential STDP kernel to spike-trains (see "Methods").

These arguments suggest that a biologically motivated learning rule naturally stores imaginary-coded information, thereby rendering it relatively resilient to the effect of homeostatically controlled synaptic fluctuations. We will next investigate how such a memory can be acquired, retained and retrieved in the presence of synaptic fluctuations. For simplicity, we will use the purely anti-symmetric $\Delta_L$.

The acquisition, i.e., the encoding and storage of a new memory trace, is initiated by stimulating the network with an external signal, $\mathbf{b}(t)$. A matrix with imaginary eigenvalues is necessarily of (at least) rank 2, corresponding to a two-dimensional space spanning the memory representation. We therefore present the network with a randomly time-varying input evolving on a plane spanned by two arbitrary directions $\mathbf{u}, \mathbf{v} \in \mathbb{R}^N$ (see "Methods"). As the input drives neural activity $\mathbf{x}$ onto the $(\mathbf{u}, \mathbf{v})$ plane, the activity-dependent learning operator $\Delta_L$ follows and becomes non-negligible, which in turn causes a change in connectivity.

The learning procedure stores geometric information of the external stimulus, specifically the directions $\mathbf{u}$ and $\mathbf{v}$, within the anti-symmetric part of the connectivity matrix. This encoding is manifested as a rank-2 operator $\mathbf{u}\mathbf{v}^\top - \mathbf{v}\mathbf{u}^\top$ which is embedded into $\mathbf{W}$. To see this, we follow the spectrum of $\mathbf{W}$ as a function of

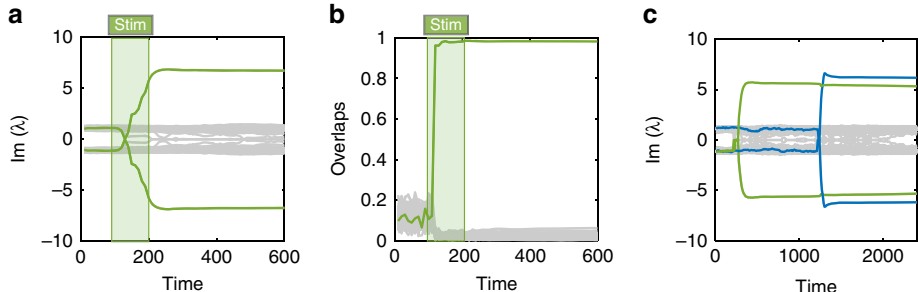

**Fig. 3** Hebbian learning by STDP embeds persistent imaginary-coded memory. **a** Imaginary part of the spectrum of $\mathbf{W}$ before, during and after external stimulation (stim.; marked by the shaded green area) applied between times $t = 100$ and $t = 200$. Before learning, the imaginary part of the spectrum is almost constant in time. The learning of a memory item manifests as the growth in imaginary amplitude of one complex conjugate eigenvalue pair (green trajectories). **b** During stimulus presentation, the learning rule modifies $\mathbf{W}$ such that the plane spanned by $\mathbf{u}$, $\mathbf{v}$ is invariant. Plotted are the overlaps of this eigenplane of $\mathbf{W}$, corresponding to the largest imaginary eigenvalue pair, with $N/2$ planes: the $\mathbf{u}$, $\mathbf{v}$ (green), and $N/2 - 1$ orthogonal planes (gray); see "Methods". **c** Zoom-out of (**a**). After the stimulus is removed, the memory representation persists (green). A second stimulus, confined to a second plane, is similarly learned at a later time (blue); both memory items are retained. This figure shows results for the decorrelation rule ($N = 128$, see "Methods"); qualitatively similar results are obtained for rate-control (see Supplementary Fig. 2)

time: during stimulus presentation, a complex conjugate eigenvalue pair forms (Fig. 3a), with corresponding eigenvectors overlapping completely with the plane spanned by $\mathbf{u}$, $\mathbf{v}$ (Fig. 3b). The strength of the memory representation—corresponding to the magnitude of the imaginary eigenvalue—depends monotonically on stimulation duration and on input amplitude. At later times, additional stimuli may be stored using the same learning protocol (Fig. 3c).

**The nature of imaginary-coded memories.** We have seen that a biologically plausible learning rule can capture the orientation in neuronal state-space of an incoming stimulus, and encode this information as a pair of imaginary eigenvalues in the network connectivity matrix. What is the nature of this memory in terms of network activity? We find that learning creates attractors in state-space, similar in fashion to those in the Hopfield model[18]. However, rather than fixed points, here the attractors are time-varying stable states—namely, limit cycles. To see this most clearly, we consider a single imaginary-coded memory embedded in the network, and examine neural dynamics while keeping $\mathbf{W}$ fixed. Following the Hopfield paradigm, we write $\mathbf{W}$ as:

$$\mathbf{W} = \rho\big(\mathbf{u}\mathbf{v}^\top - \mathbf{v}\mathbf{u}^\top\big), \tag{6}$$

where the coefficient $\rho > 0$ represents the strength of the memory representation[29].

With one stored memory as in Eq. (6), we find that, from any non-zero initial condition, the dynamics converge to periodic motion concentrated on the 'memory plane' spanned by $\mathbf{u}$ and $\mathbf{v}$. Figure 4a depicts the projections of neural activity on this plane, for two initial conditions (light gray trajectories), both converging to the limit-cycle attractor (dark closed trajectory). An approximate low-dimensional description of this limit cycle can be obtained for an infinitely steep nonlinearity $\phi$ (i.e., a step-function). The full dynamics are then well approximated by their projected coordinates on the plane, $p_\mathbf{u}$ and $p_\mathbf{v}$:

$$\begin{aligned} \dot{p}_\mathbf{u} &= -p_\mathbf{u} + \rho\, q_\mathbf{v} \\ \dot{p}_\mathbf{v} &= -p_\mathbf{v} - \rho\, q_\mathbf{u}, \end{aligned} \tag{7}$$

where $q_\mathbf{v} \approx \arctan\left(\frac{p_\mathbf{v}}{|p_\mathbf{u}|}\right)$. This low-dimensional system exhibits a stable limit-cycle around the origin (see Supplementary Note 1 and Supplementary Fig. 3). We conclude that imaginary-stored

memory items correspond to dynamic attractors, with geometry defined by that of the stimulating input. This behavior stands in contrast to the classic—symmetric—Hopfield model, where memories are represented by fixed-point attractors corresponding to fixed values of neural activity.

Embedding multiple memory planes $\{\mathbf{u}^{(k)}, \mathbf{v}^{(k)}\}_{k=1}^{M}$ corresponds to setting

$$\mathbf{W} = \mathbf{U}\mathbf{D}\mathbf{U}^\top \tag{8}$$

where the columns of $\mathbf{U}$ are the memory patterns (interleaved $\mathbf{u}^{(k)}$ and $\mathbf{v}^{(k)}$), and $\mathbf{D}$ is a $2M \times 2M$ block-diagonal matrix, with the k-th block reading $\begin{pmatrix} 0 & \rho_k \\ -\rho_k & 0 \end{pmatrix}$. Now, a locally stable limit-cycle lies on each embedded plane, and the network functions as an associative memory: initiating the dynamics within the basin of attraction of one plane—providing the network with partial information of the memory to be retrieved—leads to the recovery of the full memory item (Fig. 4b). Similar to the Hopfield model, the memory capacity is found to be proportional to system size[30]. Numerical simulations presented in Supplementary Fig. 4 show that in fact the proportionality constant is slightly higher compared with that of the symmetric Hopfield model (when normalized by a factor of two, since each memory resides on a plane; see Supplementary Note 2).

**Life cycle of a memory trace.** We next consider the entire life-cycle of a memory in the presence of synaptic fluctuations and homeostasis, starting from learning, through retention and to retrieval. During a learning event, implemented by presenting a stimulus in the two-dimensional memory plane, a memory representation is formed by the Hebbian learning rule. Figure 5a (left) shows the overlaps of neural activity onto the two planes, $r_1$ (green) and $r_2$ (blue), together with the stimulus which drives learning (shades). These projections are elevated during stimulation, which—via the Hebbian learning rule—modifies the synaptic matrix to store each of the planes in connectivity. The projection $r_3$ onto a third plane, which was not learned, is negligible as shown in the bottom line (orange).

After learning, the two memory items are stored as pairs of imaginary eigenvalues, remaining stable over time, until they are retrieved at times $t_1$ and $t_2$, respectively. At retrieval, activity is transiently attracted to the respective memory planes, as indicated by the spikes in the overlaps (Fig. 5a, right). During retrieval, activity follows the stored dynamic trajectory, exhibiting its typical oscillations (Fig. 5a, right, blue zoom). At the same time, the projection onto an arbitrary plane shows no temporal structure (orange zoom). Finally, stimulating the network with a novel cue does not elicit a significant response in neural activity in any of the projections ($t_3$ in Fig. 5a).

The effect of retrieval on the connectivity, namely on the stored memory itself, is somewhat unpredictable and depends on the exact state of the network and on the memory properties. As an example, in Fig. 5c it is seen that the green memory is damaged by retrieval, namely the magnitude of the corresponding imaginary eigenvalue is decreased. This may be caused by the homeostatic mechanism that constrains activity, in particular the component projected onto the memory plane by the retrieval event. In contrast, the blue memory is slightly strengthened by retrieval, as seen by the increased magnitude of the eigenvalue pair. In other cases the memory remains unaffected.

Throughout this entire cycle, synapses fluctuate under the effect of activity-independent noise and homeostasis. Figure 5d shows a few example synapses tracked across time, during both phases. We may disentangle the two effects, spontaneous and activity-dependent, and estimate their relative contribution to

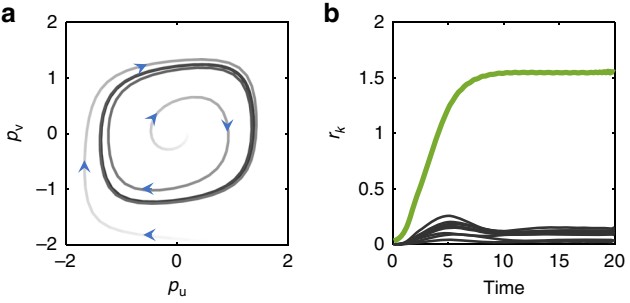

**Fig. 4** Dynamics of memory retrieval with fixed connectivity. **a** Overlaps of network activity $\mathbf{x}$ with the two directions spanning one embedded memory plane, $p_\mathbf{u} := \frac{1}{\sqrt{N}}\mathbf{u}^\top\mathbf{x}$ and $p_\mathbf{v} := \frac{1}{\sqrt{N}}\mathbf{v}^\top\mathbf{x}$ form a stable limit cycle. Shown are two trajectories, one initiated inside and the other outside the stable orbit, with arrowheads showing the direction of temporal evolution ($N = 4096$, $\rho = 4$). **b** Radial coordinates of overlaps with each of multiple ($M = 10$, $N = 4096$) embedded memory planes, $r_k := \sqrt{p_{\mathbf{u}^{(k)}}^2 + p_{\mathbf{v}^{(k)}}^2}$. Initiating the network near one plane, $\mathbf{u}^{(1)}$, $\mathbf{v}^{(1)}$, results in convergence to the associated attractor (green). Overlaps with other planes remain small (black). In this figure we construct connectivity by adding to Eq. (6) a real-coded term (see "Methods")

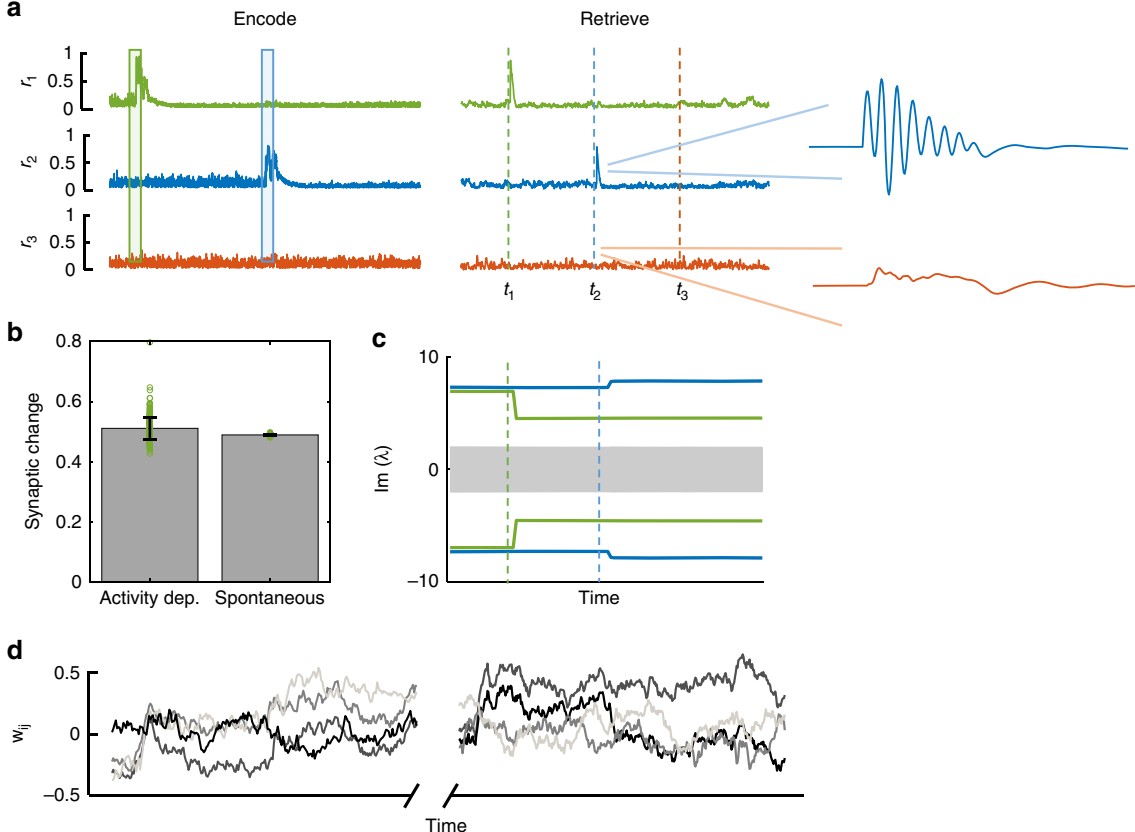

**Fig. 5** Stable memory with unstable synapses. **a** Left: Learning phase. Two distinct stimuli are sequentially presented to the network (green and blue shaded areas). Right: Retrieval phase. The embedded memories are read out of the network dynamics via a brief stimulation bearing partial information of the original stimulus. Overlaps of the network state **x** with the two learned planes are shown in corresponding color. During retrieval, activity follows the oscillatory trajectory of the dynamic memory item (blue zoom). A novel cue, on the other hand, does not elicit a significant response in neural activity (bottom trajectory, orange). **b** Relative contributions of activity-dependent (left bar) and spontaneous (right bar) synaptic fluctuations are estimated to be of similar magnitude during all phases of the memory life-cycle (see "Methods"). Error bars denote one standard deviation from the mean; overlaid data points are marked by green circles. **c** Effect of retrieval on existing memories can differ, with the green memory degrading, and the blue one strengthening. **d** Weights of four out of $N^2$ synapses (gray shades) as a function of time. The spontaneous and homeostatic contributions to plasticity drive perpetual fluctuations of synaptic weights, which occur during and after learning. This figure shows results for the rate-control homeostasis rule ($N = 128$, see "Methods")

synaptic fluctuations; their magnitude is found to be similar (Fig. 5b), in agreement with the experimentally observed phenomenon[14].

## Discussion

Experimental evidence on the perpetual changes in synaptic strengths, with and without relation to activity or learning, has accumulated by now to form a well accepted picture: synapses are not as stable as once thought. Earlier theoretic work studied the statistical properties of single synapse fluctuation using phenomenological models[12,31,32]. These models were successful in capturing the quantitative statistics of single synapses along time and across a population, but did not address their context within an active network. The more difficult question of the implications of such fluctuations to network functionality has been highlighted in several recent reviews[7–10]. Neural network models that can store memory provide a framework for addressing this problem. This can be done by adding noise and investigating its effect on memories and network dynamics.

Previous theoretical work has established the robustness of the classic Hopfield model to static synaptic noise[33,34], which exhibits a smooth and moderate decline of the critical memory load as a function of the noise variance. However, introducing a dynamic noise source to the synaptic weight evolution has a qualitatively different effect. In this case, the noise typically causes the weights to diverge with time, thus erasing any stored memory items. As shown in Fig. 2c, a dissipative term can stabilize the stochastic fluctuations, but will also erase any memory in the network. Here, we showed that it is possible to store, retain and retrieve memories in a recurrent neural network despite significant ongoing synaptic fluctuations. Motivated from a fundamental systems-theory perspective, we argued that fluctuations, and homeostasis mechanisms that control them, place a strong constraint only on the real part of the eigenvalues of the connectivity matrix. A corollary of this observation is that memories constituting stable dynamical trajectories of neural activity, associated with imaginary eigenvalues, can be kept encoded in connectivity for extended times while synapses fluctuate under noise and homeostasis.

We implemented this idea with the simplest form of such memory: a two-dimensional plane in which periodic activity persists as a stable limit cycle. This implementation extends the Hopfield picture, where memories are fixed-point attractors with static activity patterns, to memories represented by dynamic trajectories in neural state space[29]. This temporal dependence is more consistent with experimental data: the oscillatory

trajectories of memory-trace activations that arise naturally in our model resemble network-level oscillations observed during memory retrieval[35] and consolidation[36].

Since imaginary eigenvalues are associated with the anti-symmetric component of connectivity, the observed asymmetry of STDP[6] naturally suggests that memory items of this type can be learned dynamically. We have demonstrated how such learning occurs by a single stimulus presentation which spans a two-dimensional plane in activity space, thus allowing the embedding of periodic motion. This is most simply shown for a perfectly anti-symmetric STDP kernel, but is valid as long as the kernel contains a significant anti-symmetric component. The effects of symmetry of STDP on memory retention have also been noted in a different modeling context. In[23] the Hopfield model was studied in the presence of ongoing STDP; it was found that unstructured noise inserted into the neural state could stabilize memories with anti-symmetric, but not with symmetric, learning.

From a more general perspective, any learning rule represents the interaction of the system with its environment; if this rule is not homogeneous in space and time, the signature of this interaction might be encoded in some sub-space of network connectivity—a component with particular symmetries. This would allow an invariant subspace of connectivity within which memories are stored, and which is minimally tampered by homeostatic fluctuations. Indications of such invariant features have been recently observed experimentally[37]: while individual neurons exhibited significant change in their activity patterns relative to behavior in a decision task, population activity and behavior remain stable over weeks. The notion of an invariant subspace has also been suggested to underlie stable behavior during working memory tasks, despite ongoing neural activity[38].

It is also possible that other principles can be formulated that allow coexistence of stable function with synaptic fluctuations. For example, the microscopic degeneracy of representation was shown to support stable input–output relations in a feed-forward network amid strong fluctuations[39]. Although the model and its implementation are very different from ours, the motivating question is broadly similar. More recently it was proposed that in balanced cortical networks, inhibitory connectivity alone bears the burden of robust information storage[40], thereby rendering memories insensitive to fluctuations of excitatory synapses.

Our model was based on general considerations of system stability, without relying on specific implementations of homeostasis. Nevertheless, it gives rise to two points worthy of discussion in relation to experimental predictions. First, imaginary-coded memories give rise to limit-cycle attractors. Thus, retrieval of an item from long-term memory to working memory should give rise to oscillatory activity. These signatures of oscillations might be detected from the spectral properties of neural activity, expected to vary significantly between learning and rest phases (see Supplementary Note 3 and Supplementary Fig. 5). Such signatures have already been observed[41,42] but our model suggests an additional feature that may be hiding in the data. The magnitude of the imaginary eigenvalue should correlate with both the oscillation frequency and with memory strength. Indeed, for a given memory item in our model, a spectral analysis correlates almost perfectly with memory strength. Considering many different memory items, we find that, despite inter-item variability of this slope, the two measurable quantities maintain a high correlation (see Supplementary Note 3 and Supplementary Fig. 5).

Our model could, in principle, be tested by directly examining synaptic strengths during learning. The model predicts that learning-related plasticity should be preferentially anti-symmetric. This could be checked by monitoring the synaptic strengths between reciprocally connected neurons during learning and rest, and our model predicts measurable differences in these phases.

The exact correlation structure, however, is not expected to be universal and could reflect details of the homeostatic mechanisms (see Supplementary Note 3 and Supplementary Fig. 6).

More generally, our results suggest that much systems-level understanding can be gleaned by measuring and analyzing a population of synaptic strengths across time in large networks. Specifically, beyond the statistical analysis of the single synapse, invariant structure in the high-dimensional space of connectivity should be searched. Moving towards such an understanding will hopefully be possible with the advancement of experimental techniques, that will allow monitoring of multiple synapses across extended times and during various phases of behavior.

## Methods

**Model simulation.** We use home-made MATLAB software in order to numerically simulate Eqs. (1) and (2). The spectra of matrices are computed using built-in MATLAB functions, and their time-series sorted using the eigenshuffle.m MATLAB script by John D'Errico, freely available online. For the nonlinearity in firing rates, we use the hyperbolic tangent function, $\phi(z) = \tanh(z)$. We have verified that the results presented in Fig. 2 are reproduced also with a rectified-linear input–output function, $\phi(z) = \max(-5, z)$, where a negative threshold is required for achieving homeostasis. For all figures we simulate a network with $N = 128$ neurons, unless otherwise specified. Simulations with larger networks similarly exhibit all of the discussed phenomena. For numerical integration we use a time constant $dt = 0.1$. Synaptic weights were evolved with a plasticity rate $\eta = 0.01$, this includes learning and homeostatic plasticity rules. For the low-pass filter $\mathbf{y}$, we use a first-order filter with timescale $\tau = 50$ (see next "Methods" section). For the learning process we use the time-dependent input $\mathbf{b}(t) = c_{\mathbf{u}}(t)\mathbf{u} + c_{\mathbf{v}}(t)\mathbf{v}$ with $u_i, v_i \sim \mathcal{N}\left(0, \frac{1}{N}\right)$ and independent. The time-dependent functions $c_{\mathbf{u}}(t)$ and $c_{\mathbf{v}}(t)$ each follow an independent Ornstein-Uhlenbeck process with timescale 0.01. For retrieval, a brief (2 simulation time constants) pulse in the direction of $\mathbf{u}$ is applied to the network, namely $c_{\mathbf{u}}(t) = 10$ and $c_{\mathbf{v}}(t) = 0$.

For the rate-control homeostasis rule, we draw each component of the target-rate vector $\phi_0$ independently from a uniform distribution over the interval $[-1, 1]$. For the decorrelation homeostasis rule, we use $\phi_{\text{pre}}(\mathbf{x}) = \phi(\mathbf{x}) = \tanh(\mathbf{x})$ and $\phi_{\text{post}}(\mathbf{x}) = \tanh(\mathbf{x} - \bar{\mathbf{x}})$, where $\bar{\mathbf{x}}$ is a first-order low-passed version of $\mathbf{x}$, with timescale $\tau_x = 20$. In all cases, we model spontaneous fluctuations by a white noise process, $\xi_{ij}(t) \sim \mathcal{N}\left(0, \frac{1}{N}\right)$, independent across time and synapses $ij$. For the dissipative synaptic dynamics we use $\beta = 0.1$.

In Fig. 3b, we compute the overlap between two planes, spanned by $\mathbf{u}_1, \mathbf{v}_1$ and $\mathbf{u}_2, \mathbf{v}_2$ respectively, as follows: Compute the projection of each spanning vector of the first plane onto the second plane:

$$r_{\mathbf{u}_1} = \sqrt{(\mathbf{u}_1 \cdot \mathbf{u}_2)^2 + (\mathbf{u}_1 \cdot \mathbf{v}_2)^2}, \quad r_{\mathbf{v}_1} = \sqrt{(\mathbf{v}_1 \cdot \mathbf{u}_2)^2 + (\mathbf{v}_1 \cdot \mathbf{v}_2)^2},$$

and then the overlap is given by

$$\text{overlap} = \sqrt{r_{\mathbf{u}_1}^2 + r_{\mathbf{v}_1}^2}.$$

In Fig. 4, we slightly modified the imaginary-coded memory representation Eq. (6). In particular, we set connectivity to $\mathbf{W} = \rho(\mathbf{u}\mathbf{v}^\top - \mathbf{v}\mathbf{u}^\top) + \gamma(\mathbf{u}\mathbf{u}^\top + \mathbf{v}\mathbf{v}^\top)$, with $\gamma > 1$. The second term emerges naturally in $\mathbf{W}$ when the memory is learned via our dynamic learning protocol; without it, the origin in phase-space $\mathbb{R}^N$ becomes a locally stable fixed point and trajectories decay. Numerically, in the limit of small integration step $dt \to 0$, we find that, for $\gamma = 0$, the origin is actually globally stable, and the memory-related limit-cycle disappears. On the other hand for discrete-time dynamics, the model with $\gamma = 0$ is stable, and this is the version used for the capacity calculations.

For generating Fig. 5b we compute the contribution of each plasticity term $\Delta(t)$ as the temporal average of $\frac{1}{N^2}\sum_{ij}|\Delta_{ij}(t)|$, from a simulation of our model with the decorrelation homeostasis rule.

**Derivation of the Hebbian learning rule.** In this section we derive the rate-based learning rule $\Delta_L$. Our starting point is a Poisson spiking neuron with output spiking activity given by a train of point events $S_i(t) = \sum \delta(t - t_k)$ that follow from a time-dependent firing rate $\phi_i(t)$[43] with $\delta(t)$ the Dirac delta function. STDP learning is characterized by a differential update of the synaptic efficacy $W_{ij}$, based on the temporal distance $\Delta t$ between spiking of unit $i$ and unit $j$; the amplitude of change is given by the 'learning window', or kernel, $K(\Delta t)$[28]. In our rate model we are interested in variations slow compared to inter-spike intervals, and therefore we average the effect of STDP over this timescale. The average quantity which describes the temporal relation between inbound and outbound neural activity is the correlation function,

$$C_{ij}(t; t + \Delta t) = \overline{\langle S_i(t)S_j(t + \Delta t)\rangle},$$

where angular brackets denote ensemble averaging over the spiking activity and

overbar denotes temporal averaging. STDP learning can then be formalized as the convolution of this correlation function with the kernel[23]:

$$[\Delta_L]_{ij} = \eta \int_{-\infty}^{\top} ds K(t-s)C_{ij}(t;s)$$
$$+ \eta \int_{-\infty}^{\top} ds K(s-t)C_{ij}(s;t) \tag{9}$$

Within our slow-timescale approximation, correlations arise from local changes in firing rate rather than from differences between individual spikes. Therefore, in terms of the instantaneous firing rates, $C_{ij}(t;s) \approx \phi_i(t)\phi_j(s)$[43]. Using a learning kernel of the form

$$K(\Delta t) = \begin{cases} a_P e^{-\Delta t/\tau_P} & \Delta t > 0 \\ a_D e^{\Delta t/\tau_D} & \Delta t \leq 0 \end{cases}$$

where $\tau_P$, $\tau_D$ are the two timescales of the kernel, and $a_P > 0$ and $a_D < 0$ are the positive and negative amplitude for increase or decrease of the connectivity, depending on the sign of correlation, we carry out the integration in Eq. (9) to obtain

$$[\Delta_L]_{ij} = \eta\Big(a_P\phi_i(t)y_j^P(t) - a_D\phi_j(t)y_i^D(t)\Big).$$

where $y^P$, $y^D$ are first-order low-pass filters of spiking rates $\phi$, with respective timescales $\tau_P$ and $\tau_D$.

In general, the parameters of $K$ give rise to an asymmetric learning operator $\Delta_L$. The extent of asymmetry is determined by the discrepancy between the two pairs of kernel parameters, i.e. the difference in timescales of potentiation and depression $\tau_P$, $\tau_D$, and the two amplitudes $a^P$, $a^D$. When $\tau_D = \tau_P$ and $a_D = -a_P$, the learning operator is purely anti-symmetric:

$$[\Delta_L]_{ij} = \eta a_P\Big(\phi_i y_j - \phi_j y_i\Big).$$

**Reporting summary**. Further information on research design is available in the Nature Research Reporting Summary linked to this article.

## Code availability

Example code for simulating our main results can be found at https://github.com/lsusman/stable-memory.

## Data availability

Data sharing not applicable to this article as no datasets were generated or analysed during the current study.

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

## Acknowledgements

This work was supported in part by the Israeli Science Foundation (grant number 346/16, O.B.; and grant number 155/18, N.B.). We thank Noam Ziv and Lukas Geyrhofer for helpful comments on an earlier version of this manuscript.

## Author contributions

All authors conceived the model. L.S. performed all the simulations and computations. O.B. and N.B. supervised the project. All authors wrote the manuscript.

## Competing interests

The authors declare no competing interests.
