## [Peer Review File · Nature Communications]

Reviewers' Comments:

Reviewer #1:

Remarks to the Author:

This manuscript addresses an interesting and important question: how can memories be maintained in by synapses given the substantial activity-independent dynamics of synaptic connections. The solution, according to this manuscript, is that learning and fluctuations operate in independent subspaces of the connectivity matrix. Specifically, they posit that learning is restricted to the asymmetric part of the connectivity matrix, whereas activity-independent changes are restricted to the symmetric part of the connectivity matrix. This allows the network to maintain information in the connectivity while the individual synapses remain dynamic. Why should learning be restricted to the asymmetric part of the connectivity whereas fluctuations to the symmetric part? In the Methods section they derive the symmetric learning from the shape of the STDP function. Similar considerations yield the symmetric fluctuations.

Let me start with the bottom line. I liked the idea presented in the paper very much. It is not "more of the same". It is both interesting and novel. However, I find its presentation imprecise and overstated, and its relation to the experimental data lacking, even misleading, as described below:

The fluctuations model:

According to the summary of the experimental findings in p. 2, "These fluctuations are independent of neural activity". By contrast, in the fluctuations equation of p. 5 that models these fluctuations, the change in the efficacy of a synapse is a function of the product of presynaptic and postsynaptic activities. The statement and its model clearly contradict.

A related issue is that previous studies have characterized synaptic fluctuations using different quantitative models, one of them by a co-authors of this manuscript (Statman et al., PCB, 2014). I am not arguing that any of these previous models should be considered as ground truth and no changes are ever allowed in one's models. However, the authors of the current manuscript should at the very least relate their current model to those previous models, which were based on experimental measurements. They should be very clear about which aspects of their model are consistent with these previous findings and models and no less important, which are not.

I am also confused about the relationship between STDP and the presented learning rule. In the Methods section, STDP is used to derive a learning rule that differs from the one that is eventually used, on the grounds that "for the first-order filter arising from the choice of K above, a phase response of $-\pi/2$ is achieved only in the limit of an infinite frequency, the regime in which the output amplitude is close to zero. In order to ameliorate the learning procedure, we therefore construct ΔL by choosing for γ a second-order filter defined by...". I do not understand what they mean by the word "ameliorate". If we believe that the standard STDP rule describes the physical reality of synapses, then it should not be "ameliorated".

Finally, I am having a hard time applying their framework to a more realistic setting, in which not all neurons are connected to all neurons. Please allow me to explain. The STDP rule describes how neural activity changes the strength of connection between already-connected neurons. In any cortical area, connectivity is partial and most neurons are not connected to most neurons. Correct me if I am wrong, but if neuron A is connected to neuron B but neuron B is not connected to neuron A, the learning and fluctuation components of the synaptic plasticity rules cannot be anti-symmetric and symmetric, respectively, as posited in this manuscript.

Having made all these points, I am not arguing that the authors should change their model or that a

failure to address these points renders the model useless. "All models are wrong". However, the inconsistencies between this model and previous models, as well as the physical reality, should be stated in a clear and honest way.

Presentation of results:

The quantitative results are presented in the text in a sloppy, impressionist way. While it is useful to provide some insight, I remain at loss when attempting to understand the facts. I will name a two examples, but the Results section should be rewritten and the authors should be very precise in their statements, and make it very clear which results are based on mathematical proofs (and what are the conditions), which are based on numerical simulations and which are merely speculations.

This issue is particularly problematic when the capacity of the model is discussed. In the main text, p. 14, the connectivity matrix following the embedding of multiple memory patterns is presented. Shortly afterwards it is stated in p. 15 that "stable heterogeneous mixtures exist for any memory load $\alpha \leq 0.5$ ". I was led to believe that this is a general and exact result. Looking for the "proof", I learn from section S5, assuming that I understood it correctly, that (1) it is assumed that the patterns are orthogonal and not general (and with equal norms?); (2) the result is based on numerical simulations. Moreover, considering these simulations, certainly in the continuous case (Fig. S6), there is no qualitative change in behavior when $\alpha = 0.5$.

Another example of this impreciseness is that the authors claim in several places that the dynamics are chaotic. This claim is not supported by any mathematical proof or even numerical simulations, rather than showing examples of what seems like irregular activity in figs and movie. I am not arguing that the dynamics are not chaotic. Nor am I arguing that this is a crucial point in the manuscript. However, if they make this claim then they should support it.

Additional point:

To what extent do the results depend on the shape of the non-linearity (tanh)?

Minor:

p.8, "Irregular modulations" – clarify

Order of authors in Ref. 15 is wrong and there is an extra WB.

Fig. S6 is denoted as S5 in legend.

Reviewer #2:

Remarks to the Author:

A long standing dogma in neuroscience is that memories are stored through synaptic modifications. While many experiments support this hypothesis, other experiments have shown that synapses can also exhibit strong fluctuations that do not seem to be related to learning. One important question is then how can networks maintain stable memories in the face of such synaptic volatility. This is the question addressed by Susman and colleagues. They hypothesize that two types of dynamics operate on two components of the connectivity matrix

- a Hebbian dynamics modifying the antisymmetric component of the connectivity matrix (due to the temporal asymmetry of spike-timing dependent plasticity), and a homeostatic dynamics that act on the symmetric component of the connectivity matrix. They show that this strict separation leads to stable memory storage, in which stored memories represent limit cycles. They also present a more general argument in supplementary information showing a generalization to arbitrary decompositions in two orthogonal sub-spaces of the connectivity matrix. The issues discussed in the manuscript are certainly important, and the scenario studied by the authors is interesting, but I have two major issues with the paper.

1. Novelty: The scenario is very similar to the paper of Magnasco et al (2009) - In particular the dynamics of the symmetric component in the weight is the same as in Magnasco et al, who already noted that with these dynamics the anti-symmetric component would be unaffected by such dynamics and therefore memories stored through STDP would be stable.

2. Biological realism: The paper claims that the dynamics of the symmetric component represents a 'mathematically consistent implementation of known homeostatic mechanisms' using a purely symmetric dynamics, but I did not find any justification for this statement. There is no 'known homeostatic mechanism' (as far as I know) that would be constrained to be in the symmetric sub-space. Known homeostatic mechanisms depend purely on the post-synaptic neuron, not on the pre-synaptic one - the goal being to maintain post-synaptic firing rate constant. Such a mechanism will definitely NOT be constrained to the symmetric sub-space.

The model predicts that the learned part of the connectivity should be anti-symmetric. The only data I know relating statistics of connectivity to function are the studies by Mrsic-Flogel and collaborators, that found that pairs of neurons with similar orientations have increased bidirectional synaptic connections. This should lead to a significant symmetric component of the connectivity matrix (consistent with a 'ring model'), at odds with this prediction.

In addition, the characterization of STDP as potentially a purely anti-symmetric rule is likely to be overly simplistic - work by many authors have shown that the picture is much more complex and spike timing is only one of the many factors affecting synaptic plasticity. For instance work by Sjostrom et al (2001) showed that at high pre and post synaptic rates plasticity synapses potentiate regardless of timing, while at low pre and post rates they depress, again regardless of timing. Graupner et al (2016) argue that synaptic efficacy is likely to be affected much more strongly by rate than by timing, which would potentially lead to a predominantly symmetric, not antisymmetric, rule.

All these elements seem to point in a direction that is completely

opposite to the one proposed by the authors - a predominantly symmetric Hebbian plasticity, together with an approximately asymmetric homeostatic dynamics. It is unclear whether the scenario proposed by the authors would survive in this case.

Minor issues:

- p.4 ref.[22]: that paper did not use a rate model.
- homeostatic plasticity being symmetric in time and ref.[26]: It seems the justification for relating homeostasis and symmetry comes from the temporally symmetric plasticity rule used in that paper (but first proposed by Vogels et al in their Science 2011 paper). However, this plasticity rule acts on inhibitory to excitatory synapses (not excitatory to excitatory) - a type of connectivity that is constrained to be antisymmetric, not symmetric. So it can certainly not serve as a justification for the rule used the authors.
- p.6, line 2: the authors probably refer to Fig.3B not Fig.2B.
- the authors make a big deal in introduction of the fact the magnitude of the 'spontaneous' changes are expected to be of the same order as the ones due to Hebbian learning, but they actually never show that this is the case in their model.
- the assumption of orthogonal patterns is particularly unrealistic. What happens when one relaxes this assumption (i.e. with random patterns as in the classical Hopfield model)?
- the reference list needs to be cleaned up quite a bit (what is 's.l.'? volume and page numbers missing; scrambled names; etc)
- the curves shown in Fig.S5 and 6 are very noisy. Is this from a single realization? The authors should show averages (and SD or SE) over many realizations.

Reviewer #3:

Remarks to the Author:

In the paper "Stable memory with unstable synapses" by Susman et al the authors study the dynamics of neuronal networks subject to synaptic plasticity that is driven both by learning and random fluctuations that induce changes of the same order as the plastic changes. By separating synaptic connectivity dynamics into a symmetric fluctuating part and a anti-symmetric plasticity part the network is able to store and recover memories as the fluctuations do not affect the anti-symmetric sub-space. The encoding of memories in the anti-symmetric sub-space is motivated from reducing STDP to firing rate models in which perfectly anti-symmetric STDP leads to anti-symmetric modification dynamics of the synaptic weights.

The study is interesting and mostly clearly written (see detailed comments below). The math is interesting and mostly correct in my view (also see comments below)

On the critical side, the potential relevance to biology could be discussed more thoroughly, particularly in the light of several recent biological studies (including "Locally coordinated synaptic plasticity of visual cortex neurons in vivo", by E-Boustani et al, Science 2018, "Dynamic Reorganization of Neuronal Activity Patterns in Parietal Cortex", Driscoll et al. Cell 2018 (see also comments below).

The 'orthogonality in synaptic weight space' is not necessarily new (see e.g. "A theory for how sensorimotor skills are learned and retained in noisy and non-stationary neural circuits", Ajemain et al., PNAS 2013), so the authors might want to clarify the differences in their approach.

Constant turn-over of synapses is costly, thus having synaptic connectivity fluctuations probably play other interesting roles.

While the authors approach to explaining stable memories is justified, some additional focus on what the fluctuations may be useful for and particularly what roles they can play in the proposed model by the authors might be useful. The authors mention aspects of the fluctuations of their 'more dynamical approach' in the discussion. In my view, this is a very interesting aspect that could be elaborated on in more detail and clarity, even already in a results section.

In summary, I think the manuscript is interesting and could be considered further if appropriately revised. I leave it to the editor to decide if it meets the criteria of broad enough interest for publication in Nature Communications.

General comments:

1.) Robustness:

The authors consider networks in which all neurons can make potential connections to all others. Typically connectivity is constrained and often in an asymmetric manner (e.g. neuron a that projects to neuron b does not necessarily have access to inputs from neuron b). It might be useful to see how robust the proposed mechanisms are against such constraints.

The asymmetry might be even more pronounced taking into account Dale's principle and separation of inhibitory vs excitatory neurons).

2.) Assumptions:

a.) The derivation from STDP rules to the asymmetric vector field of the weight changes appears to make several assumptions. The manuscript could benefit strongly from making those more clear and also discussing them in the main text. E.g. the correlations are approximated as the product of firing rates, when is this valid? There are several averaging procedures used, what are the time scales for those, and what conditions make the firing rate approach valid? Does a full spiking neuronal network reproduce the overall results?

b.) The low pass filtering that arises in the reduction process from STDP to the firing rate model is further modified to a second order filtering with a resonance to achieve a $\pi/2$ phase shift and make the anti-symmetric learning more effective. This property while seemingly crucial is not well discussed in the main manuscript. In fact, it is unclear to me how crucial this modification is. Moreover, is such a 'resonance' in the learning rule biologically plausible or is there any experimental evidence?

3.) Mechanism:

a.) Acquisition: From the presentation of the manuscript it is not fully clear how the stimuli are presented. In the methods 'circular' activation in the stimulus plane and a random walk is mentioned

but how do those compare for the learning and resulting memory limit cycle ? What happens for higher dim stimuli ?

b.) Memory bound: 2 dimensions are used to encode memory which would make one think to find a memory bound α that is smaller than the Hopfield bound ? Why is it still 1/2 ?

4.) Clarity

In general - as the manuscript is theoretical work- it could benefit by (i) being more precise in the main statements,

(ii) integrating more information about the crucial derivation steps into the main flow of the manuscript.

Detailed comments:

Introduction:

As also mentioned earlier, there are several computational studies on 'invariant subspaces' that might deserve to be discussed initially or in the discussion part. (e.g. Neuronal circuits underlying persistent representations despite time varying activity" Druckmann et al, and several others).

Results

Paragraph below eq (2)

This paragraph could be more clear, stating also all the assumptions that go into the derivation. Most importantly, clarifying how the filtered version y arises in the derivation, how it is modified to a second order filter, and what this means for the learning rule, would be beneficial in my view.

Last paragraph on page 5

Similar as in the last comment also more details and justification on the symmetric term could be helpful for the reader. E.g. some more details and intuition on the role of the $\phi \phi^T$ homeostasis terms could be useful. Particularly as in earlier references to S1-1 this term is already used but in a non-symmetric way. It might be worth starting introducing these earlier so when jumping to the supplement things are already clear.

Figure 3.

Axes have no scales, but this Figure is referred to later on page 16 asking the reader to compare scales !

Figure 4.

A,C, After learning there is a ringing in the Im part of the relevant eigenvalue (also visible in other figures). Where is this ringing coming from ?

The can can be more precise, particularly how the the overlaps in B are calculated should be more explicitly stated.

D/ Why does the suppression of the memory saturate between 0.2 and 0.3 ?

Section 2.2. Retention

In my view, how the attraction of the neuronal activity to the memory plane is created could be explained in more detail.

Page 11

As already stated above, the homeostasis term $f(x)g(x)^T$ should be better motivated, how could this term arise biologically?

Figure 5.

Is there a way to understand how large the asymmetry in the homeostasis term has to be in order to ensure stability ?

After eq. (5).

Sufficiently large ρ should be made more specific. Also ρ is related to the imprinting mechanism as stated on page 9.

It would be good to understand the relation between stimulus duration and amplitude and ρ more precisely, as this seems to determine whether the memory becomes a limit cycle.

Figure 6.)

The figure and its caption on its own is confusing as different conditions lead to the different panels. Those should be stated clearly in the caption. The magnitudes in the various panels of the memory patterns differ, are of the order of $O(100)$ in A,B and $O(2)$ in C.

Caption A. Is a square root missing over the N in the coordinates p_u, p_v ?

Panel B. What happens if the overlap initially is much smaller? Further, what are the mechanisms that stabilize certain memory components apart from the main one while others are continuing to decay ?

Panel C is confusing together with its caption in my view, it might be easier to explicitly show time on one axis by reducing the p coordinates to a radius as in B, or use a better time coding and reproduce that in D ?

How do these simulations in Fig. compare to the reduced dynamics / theoretical predictions from the SI?

Page 15. Memory load: 2 dimensions are used to encode memory which would make one expect to find a memory bound α that is half the Hopfield bound ? Why is it still M/N ?

Page 15. Ref to Fig 3B is either wrong or not appropriate as Fig 3 does not have any scales to compare to.

Page 15, last paragraph before Discussion

The results here are quite interesting but would need a more thorough explanation in my view, same for SI-6.

Discussion

The produced memory traces show oscillatory activity in a single pair of neurons in this model while neuronal oscillations typically require a much larger set of neurons. In my understanding the population activity of the model network studied here is not coherent on the population level and thus would not show any macroscopic oscillations. Further the oscillation frequency is related to the memory strength in my view in this model ? In my view the relation to the experimental studies is therefore too speculative. Do the experimentally observed oscillations really reflect the dynamics observed in this model ?

Maybe the results on the 'limit cycle memories' generalize to larger dimensional stimuli and larger dimensional memory spaces, but would they still for simple limit cycles there or produce a much more complex dynamical state ?

As pointed out in the intro to this report, the idea of orthogonality is not necessarily novel and in my view needs more discussion.

Making a connection to an experimental prediction is quite interesting and it might be beneficial to the manuscript to move Figure S6 to the main text.

SI-S1

- $R^{N \times N}$ is not isomorphic to $R^{N^2 \times 1}$ but R^{N^2} only !

- S1 is referenced in the main text to before explaining much of the notation used here. It might be good to either explain it in S1 or reference to S1 later in the main manuscript.

SI-S2

The video needs more explanation, it is unclear what network is precisely simulated etc.

SI-Figure S3.

A few more data points on how the resonance frequency mismatch affects learning could be interesting, See also remarks to the results section above.

SI-S4

Derivation seems correct, but W is always wrongly defined as zero in the SI while I think $uv^T - vu^T$ is meant.

SI-S5

The link to the 4-cycle result is nice.

Figure S5: It is not clear to me how the figure is generated. Are the results averages of many networks and initial states ?

Where are the errors and how big are they on the curves ?

Why is the L1 radius used and not the L2 radius for m ? What is the most comparable overlap measure between the classical Hopfield network and the one used here?

Why is there so much more variance in the anti-symmetric model in the curves in general ?

As pointed out above, why is the limiting capacity at $\alpha = 1/2$ and not some smaller value as always two dimensions are used to store the memory ?

Figure S5-2

It seems the anti-symmetric part in A and B performs worse (and the curves are much more fluctuating in comparison to the symmetric Hopfield network. The text states otherwise ?

Again no error bars and info on how the curves are obtained is given.

SI-S6

Given the definition of $q \sim 1/\sqrt{N}$ in SI-S4, there is a factor of N missing in my opinion in the dynamics for δ and γ in front of the q and in the following equations ?

SI-S7

The model predictions are interesting. Is there any way to get some analytic handle on those results ?

References

Several references seem to be wrongly cited or missing the last author.

Reviewer #1 (Remarks to the Author):

This manuscript addresses an interesting and important question: how can memories be maintained in by synapses given the substantial activity-independent dynamics of synaptic connections. The solution, according to this manuscript, is that learning and fluctuations operate in independent subspaces of the connectivity matrix. Specifically, they posit that learning is restricted to the asymmetric part of the connectivity matrix, whereas activity-independent changes are restricted to the symmetric part of the connectivity matrix. This allows the network to maintain information in the connectivity while the individual synapses remain dynamic. Why should learning be restricted to the asymmetric part of the connectivity whereas fluctuations to the symmetric part? In the Methods section they derive the symmetric learning from the shape of the STDP function. Similar considerations yield the symmetric fluctuations.

Let me start with the bottom line. I liked the idea presented in the paper very much. It is not "more of the same". It is both interesting and novel. However, I find its presentation imprecise and overstated, and its relation to the experimental data lacking, even misleading, as described below:

The fluctuations model:

According to the summary of the experimental findings in p. 2, "These fluctuations are independent of neural activity". By contrast, in the fluctuations equation of p. 5 that models these fluctuations, the change in the efficacy of a synapse is a function of the product of presynaptic and postsynaptic activities. The statement and its model clearly contradict.

This is true. Stated more precisely, the experimental results do not show that fluctuations are independent of neural activity, but that they persist in the absence of activity. In fact the distribution broadens in the absence of activity, pointing to its role in homeostatic restraint, as described mathematically in our model. In the revised version we include also a truly activity-independent noise term.

We thank the reviewer for sharpening this delicate point and have modified the text to faithfully align with the experimental results.

A related issue is that previous studies have characterized synaptic fluctuations using different quantitative models, one of them by a co-authors of this manuscript (Statman et al., PCB, 2014). I am not arguing that any of these previous models should be considered as ground truth and no changes are ever allowed in one's models. However, the authors of the current manuscript should at the very least relate their current model to those previous models, which were based on experimental measurements. They should be very clear about which aspects of their model are consistent with these previous findings and models and no less important, which are not.

Previous modeling efforts were focused on the statistics of fluctuations in individual synapses. These single-synapse statistics were well described by a simple one-dimensional model – which, however, disregarded the surrounding network, its neural activity and the relationship between the two. In the current work we neglect the quantitative statistics of single synapses and study the implication of ANY fluctuations on the broader system of the network and its functionality, taking into account the above mentioned relations between synaptic dynamics and neuronal activity. For this reason the fluctuation model is simpler in itself and synaptic distributions are Gaussian. A goal for future work is to reconcile all these aspects of synaptic fluctuations in a single framework.

This comparison with previous work is now described in the paper. We thank the reviewer for helping us improve the context of the current work.

I am also confused about the relationship between STDP and the presented learning rule. In the Methods section, STDP is used to derive a learning rule that differs from the one that is eventually used, on the grounds that "for the first-order filter arising from the choice of K above, a phase response of $-\pi/2$ is achieved only in the limit of an infinite frequency, the regime in which the output amplitude is close to zero. In order to ameliorate the learning procedure, we therefore construct ΔZ by choosing for γ a second-order filter defined by...". I do not understand what they mean by the word "ameliorate". If we believe that the standard STDP rule describes the physical reality of synapses, then it should not be "ameliorated".

This problem has now been resolved, since in the revised manuscript we use a first order filter. None of the results are qualitatively affected by this change. Indeed the relation to standard STDP is now more transparent.

Finally, I am having a hard time applying their framework to a more realistic setting, in which not all neurons are connected to all neurons. Please allow me to explain. The STDP rule describes how neural activity changes the strength of connection between already-connected neurons. In any cortical area, connectivity is partial and most neurons are not connected to most neurons. Correct me if I am wrong, but if neuron A is connected to neuron B but neuron B is not connected to

neuron A, the learning and fluctuation components of the synaptic plasticity rules cannot be anti-symmetric and symmetric, respectively, as posited in this manuscript.

Following this remark, we have tested whether our learning procedure works also for sparse connectivity matrices. While it is true that in this case the symmetric and antisymmetric parts of the matrix are not biologically relevant quantities, the symmetries of the learning rules still have meaning. For example, STDP cannot strengthen both directions of a connection simultaneously. If these learning rules act while maintaining sparse connectivity, (e.g. act on existing connections), all the results presented in the paper are qualitatively conserved. Fig. 1 below demonstrates that an embedded imaginary-coded memory item is stably maintained despite fluctuations and homeostasis, also in a sparse network.

Fig. 1: Extension of results to sparse networks. We validate the robustness of imaginary-coded memory in the case of sparse connectivity. We simulate a network with $N = 128$ neurons, synapses fluctuate randomly as in the main text, and are subject to rate-control homeostasis. The synaptic matrix is subject to a topological constraint – the non-zero connections are randomly chosen with probability $p = 0.2$ and are constant across a simulation.

Having made all these points, I am not arguing that the authors should change their model or that a failure to address these points renders the model useless. "All models are wrong". However, the inconsistencies between this model and previous models, as well as the physical reality, should be stated in a clear and honest way.

Presentation of results:

The quantitative results are presented in the text in a sloppy, impressionist way. While it is useful to provide some insight, I remain at loss when attempting to understand the facts. I will name two examples, but the Results section should be rewritten and the authors should be very precise in their statements, and make it very clear which results are based on mathematical proofs (and what are the conditions), which are based on numerical simulations and which are merely speculations.

This issue is particularly problematic when the capacity of the model is discussed. In the main text, p. 14, the connectivity matrix following the embedding of multiple memory patterns is presented. Shortly afterwards it is stated in p. 15 that "stable heterogeneous mixtures exist for any memory load $\alpha \leq 0.5$ ". I was led to believe that this is a general and exact result. Looking for the "proof", I learn from section S5, assuming that I understood it correctly, that (1) it is assumed that the patterns are orthogonal and not general (and with equal norms?); (2) the result is based on numerical simulations. Moreover, considering these simulations, certainly in the continuous case (Fig. S6), there is no qualitative change in behavior when $\alpha=0.5$.

Following this general remark, we have re-written the Results section and made every effort to describe our analysis and results as concisely as possible. We furthermore answer to the two specific issues raised in this context:

Specifically, the issue of capacity is now elaborated in a separate Supplementary chapter, including the scaling of capacity with network size, comparison to the known properties of the Hopfield model, and a clear explanation that these are numerical results.

Another example of this impreciseness is that the authors claim in several places that the dynamics are chaotic. This claim is not supported by any mathematical proof or even numerical simulations, rather than showing examples of what seems like irregular activity in figs and movie. I am not arguing that the dynamics are not chaotic. Nor am I arguing that this is a crucial point in the manuscript. However, if they make this claim then they should support it.

The chaotic nature of the dynamics (with noise and homeostasis in synapses) is supported by computation of the maximal Lyapunov exponent, for several values of the fluctuation parameter η_F (Fig. 2 below). However, since this point is not crucial for our purposes, we refrain from discussing it.

Figure 2: numerical computation of the maximal Lyapunov exponent for the model with decorrelating homeostasis. For the values of η_F tested, between 10^{-3} and 10^{-1} , this exponent is positive, implying chaotic dynamics.

Additional point:

To what extent do the results depend on the shape of the non-linearity (\tanh)?

We have tested the robustness of our results with respect to the type of nonlinearity ϕ . Below is a simulation validating the memory erosion properties presented in the main text, for the two homeostatic mechanisms discussed in the revised paper, with ϕ is a rectified linear function instead of the \tanh function used in the paper.

Figure 3: Analog of Fig. 2 in the revised manuscript, with a different nonlinear function. Real (top) and imaginary (bottom) memories are embedded in the connectivity matrix, and the corresponding part of the eigenvalue spectrum is shown during evolution of Eq. (1) in the main text, with different homeostasis mechanisms (A: decorrelation, B: rate control).

Minor:

p.8, "Irregular modulations" – clarify

The combination of activity independent plasticity with homeostasis leads to chaotic fluctuations – which can be described as irregular.

This phrasing has been removed from the revised manuscript.

Order of authors in Ref. 15 is wrong and there is an extra WB.

Corrected.

Fig. S6 is denoted as S5 in legend.

Corrected.

Reviewer #2 (Remarks to the Author):

A long standing dogma in neuroscience is that memories are stored through synaptic modifications. While many experiments support this hypothesis, other experiments have shown that synapses can also exhibit strong fluctuations that do not seem to be related to learning. One important question is then how can networks maintain stable memories in the face of such synaptic volatility. This is the question addressed by Susman and colleagues. They hypothesize that two types of dynamics operate on two components of the connectivity matrix - a Hebbian dynamics modifying the antisymmetric component of the connectivity matrix (due to the temporal asymmetry of spike-timing dependent plasticity), and a homeostatic dynamics that act on the symmetric component of the connectivity matrix. They show that this strict separation leads to stable memory storage, in which stored memories represent limit cycles. They also present a more general argument in supplementary information showing a generalization to arbitrary decompositions in two orthogonal sub-spaces of the connectivity matrix. The issues discussed in the manuscript are certainly important, and the scenario studied by the authors is interesting, but I have two major issues with the paper.

1. Novelty: The scenario is very similar to the paper of Magnasco et al (2009) - In particular the dynamics of the symmetric component in the weight is the same as in Magnasco et al, who already noted that with these dynamics the anti-symmetric component would be unaffected by such dynamics and therefore memories stored through STDP would be stable.

Indeed, our study draws inspiration and is in part based on this previous work. Our work goes significantly beyond previous work in constructing a concrete memory model, which minimally interferes with homeostasis, and investigating its properties. The new modified version actually tests our general picture using also a different homeostasis mechanism, and thus extends beyond this previous work also in this sense.

2. Biological realism: The paper claims that the dynamics of the symmetric component represents a 'mathematically consistent implementation of known homeostatic mechanisms' using a purely symmetric dynamics, but I did not find any justification for this statement. There is no 'known homeostatic mechanism' (as far as I know) that would be constrained to be in the symmetric sub-space. Known homeostatic mechanisms depend purely on the post-synaptic neuron, not on the pre-synaptic one - the goal being to maintain post-synaptic firing rate constant. Such a mechanism will definitely NOT be constrained to the symmetric sub-space.

We agree that this claim was not well-founded in the previous version of the manuscript. Based on this remark, in the revised paper we study the effect of two homeostatic mechanisms with different symmetry properties, one of which maintains firing rate as suggested by the reviewer. We find that our main result, that imaginary-encoded memories persist despite fluctuations and homeostasis, actually follows from general considerations of system stability and does not require that the homeostasis mechanism be confined to the symmetric subspace. We thank the reviewer for this remark which enabled us to extend the validity of our results.

The model predicts that the learned part of the connectivity should be anti-symmetric. The only data I know relating statistics of connectivity to function are the studies by Mrcsic-Flogel and collaborators, that found that pairs of neurons with similar orientations have increased bidirectional synaptic connections. This should lead to a significant symmetric component of the connectivity matrix (consistent with a 'ring model'), at odds with this prediction.

The results of Mrcsic-Flogel et al. show bidirectionality but not symmetry (For instance the connection between neurons 1&2 in Figure 3 of that paper has a very strong asymmetry). The statistics of symmetry cannot be deduced from their data due to small sample sizes (as explained by the authors). These results on bidirectionality have been noted to be in apparent discrepancy with standard STDP (e.g. Miner et al., 2017; Clopath et al., 2010). They suggest that two opposing forces are at play: STDP pushing towards feed-forward directed structures, and bidirectionality pushing towards symmetry. In our model bidirectionality is promoted by a symmetric component of homeostasis. The learned part of connectivity is not strictly anti-symmetric, but has a large anti-symmetric component, in agreement with the standard view of STDP.

In addition, the characterization of STDP as potentially a purely anti-symmetric rule is likely to be overly simplistic - work by many authors have shown that the picture is much more complex and spike timing is only one of the many factors affecting synaptic plasticity. For instance work by Sjostrom et al (2001) showed that at high pre and post synaptic rates plasticity synapses potentiate regardless of timing, while at low pre and post rates they depress, again regardless of timing. Graupner et al (2016) argue that synaptic efficacy is likely to be affected much more strongly by rate than by timing, which would potentially lead to a predominantly symmetric, not antisymmetric, rule.

In the revised version, we model STDP not as purely anti-symmetric but rather as having a strong anti-symmetric component. This is in line with the currently accepted standard description of STDP. We certainly agree that this model is a simplification, and does not properly account for effects such as nonlinearities at extreme firing rates.

All these elements seem to point in a direction that is completely opposite to the one proposed by the authors - a predominantly symmetric Hebbian plasticity, together with an approximately asymmetric homeostatic dynamics. It is unclear whether the scenario proposed by the authors would survive in this case.

We can summarize our previous replies as follows: (1) in the revised paper we have modified our model so that the fluctuation and learning elements are not purely symmetric or anti-symmetric, respectively; the qualitative features of the model remain. (2) We find that currently available data favors an approximately anti-symmetric behavior of STDP, and is inconclusive with respect to the homeostatic component.

Minor issues:

- p.4 ref.[22]: that paper did not use a rate model.

Thank you for noticing this mistake, it was corrected and alternative reference provided.

- homeostatic plasticity being symmetric in time and ref.[26]: It seems the justification for relating homeostasis and symmetry comes from the temporally symmetric plasticity rule used in that paper (but first proposed by Vogels et al in their Science 2011 paper). However, this plasticity rule acts on inhibitory to excitatory synapses (not excitatory to excitatory) - a type of connectivity that is constrained to be antisymmetric, not symmetric. So it can certainly not serve as a justification for the rule used the authors.

As stated above, we no longer assume that homeostatic plasticity is symmetric.

- p.6, line 2: the authors probably refer to Fig.3B not Fig.2B.

These figures have been modified in the revised version.

- the authors make a big deal in introduction of the fact the magnitude of the 'spontaneous' changes are expected to be of the same order as the ones due to Hebbian learning, but they actually never show that this is the case in their model.

A comparison of the magnitudes of the two effects is added in Fig. 5C for the modified model version – showing that they are similar.

- the assumption of orthogonal patterns is particularly unrealistic. What happens when one relaxes this assumption (i.e. with random patterns as in the classical Hopfield model)?

In the revised paper, all patterns were chosen at random (replacing the orthogonal patterns in the previous version). The results and conclusions are not qualitatively affected.

- the reference list needs to be cleaned up quite a bit (what is 's.l.'? volume and page numbers missing; scrambled names; etc)

Cleaned. Thank you.

- the curves shown in Fig.S5 and 6 are very noisy. Is this from a single realization? The authors should show averages (and SD or SE) over many realizations.

This remark has been addressed in the modified supplementary chapter, where results were averaged over a large number of realizations.

Reviewer #3 (Remarks to the Author):

In the paper "Stable memory with unstable synapses" by Susman et al the authors study the dynamics of neuronal networks subject to synaptic plasticity that is driven both by learning and random fluctuations that induce changes of the same order as the plastic changes. By separating synaptic connectivity dynamics into a symmetric fluctuating part and a anti-symmetric plasticity part the network is able to store and recover memories as the fluctuations do not affect the anti-symmetric sub-space. The encoding of memories in the anti-symmetric sub-space is motivated from reducing STDP to firing rate models in which perfectly anti-symmetric STDP leads to anti-symmetric modification dynamics of the synaptic weights.

The study is interesting and mostly clearly written (see detailed comments below). The math is interesting and mostly correct in my view (also see comments below)

On the critical side, the potential relevance to biology could be discussed more thoroughly, particularly in the light of several recent biological studies (including "Locally coordinated synaptic plasticity of visual cortex neurons in vivo", by E-Boustani et al, Science 2018, "Dynamic Reorganization of Neuronal Activity Patterns in Parietal Cortex", Driscoll et al. Cell 2018 (see also comments below).

We strengthened the relevance to biology in two ways. First, we generalized our results to another homeostatic rule – rate control. This rule is inspired by experimental findings like those of El-boustani et al. Second, we elaborate on the possible implications of our model, and how they could be measured in future experiments.

The 'orthogonality in synaptic weight space' is not necessarily new (see e.g. "A theory for how sensorimotor skills are learned and retained in noisy and non-stationary neural circuits", Ajemain et al., PNAS 2013), so the authors might want to clarify the differences in their approach.

Thank you for pointing out this reference. We have now included a discussion of this previous work and its relation to ours in the Discussion of the revised manuscript.

Constant turn-over of synapses is costly, thus having synaptic connectivity fluctuations probably play other interesting roles. While the authors approach to explaining stable memories is justified, some additional focus on what the fluctuations may be useful for and particularly what roles they can play in the proposed model by the authors might be useful. The authors mention aspects of the fluctuations of their 'more dynamical approach' in the discussion. In my view, this is a very interesting aspect that could be elaborated on in more detail and clarity, even already in a results section.

This is an interesting remark. The role of noise in various learning algorithms is a topic of much current work. We view this as a topic for future research and remark on it in the revised manuscript.

In summary, I think the manuscript is interesting and could be considered further if appropriately revised. I leave it to the editor to decide if it meets the criteria of broad enough interest for publication in Nature Communications.

General comments:

1.) Robustness:

The authors consider networks in which all neurons can make potential connections to all others. Typically connectivity is constrained and often in an asymmetric manner (e.g. neuron a that projects to neuron b does not necessarily have access to inputs from neuron b). It might be useful to see how robust the proposed mechanisms are against such constraints.

The asymmetry might be even more pronounced taking into account Dale's principle and separation of inhibitory vs excitatory neurons).

Please see Fig. 1 above, in replies to Reviewer 1, showing the robustness of our results to sparse connectivity.

2.) Assumptions:

a.) The derivation from STDP rules to the asymmetric vector field of the weight changes appears to make several assumptions. The manuscript could benefit strongly from making those more clear and also discussing them in the main text. E.g. the correlations are approximated as the product of firing rates, when is this valid? There are several averaging procedures used, what are the time scales for those, and what conditions make the firing rate approach valid? Does a full spiking neuronal network reproduce the over all results?

This derivation is detailed in a dedicated Methods section, and most of it goes back to previous work. We provide the main steps and give reference where extended justification is required.

b.) The low pass filtering that arises in the reduction process from STDP to the firing rate model is further modified to a second order filtering with a resonance to achieve a $\pi/2$ phase shift and make the anti-symmetric learning more effective. This property while seemingly crucial is not well discussed in the main manuscript. In fact, it is unclear to me how crucial this modification is. Moreover, is such a 'resonance' in the learning rule biological plausible or is there any experimental evidence?

In the revised manuscript we use a first-order filter.

3.) Mechanism:

a.) Acquisition: From the presentation of the manuscript it is not fully clear how the stimuli are presented. In the methods 'circular' activation in the stimulus plane and a random walk is mentioned but how do those compare for the learning and resulting memory limit cycle ?

In the revised manuscript we use only random walk inputs. This is explained in the Methods section.

What happens for higher dim stimuli ?

Simulations that we performed to test this point indicate that higher dimensional stimuli store several distinct limit cycles. Our results on multiple memory elements, presented in Fig. 4B, shows that these planes act as separate attractors.

b.) Memory bound: 2 dimensions are used to encode memory which would make one think to find an memory bound α that is smaller the Hopfield bound? Why is it still 1/2 ?

This is correct. When comparing the memory capacity of our model to that of Hopfield, the comparison is made while retaining the total dimensionality of memory space the same. This point is now explained clearly in the text.

4.) Clarity

*In general - as the manuscript is theoretical work- it could benefit by (i) being more precise in the main statements,
(ii) integrating more information about the crucial derivation steps into the main flow of the manuscript.*

In the revised manuscript we tried to be more precise in stating the results and how they were obtained.

Detailed comments:

Introduction:

As also mentioned earlier, there are several computational studies on 'invariant subspaces' that might deserve to be discussed initially or in the discussion part. (e.g. Neuronal circuits underlying persistent representations despite time varying activity" Druckmann et al, and several others).

The Discussion section now contains reference and discussion of this (and other) previous work.

Results

Paragraph below eq (2)

This paragraph could be more clear, stating also all the assumptions that go into the derivation. Most importantly, clarifying how the filtered version y arises in the derivation, how it is modified to a second order filter, and what this means for a the learning rule, would be beneficial in my view.

This paragraph is removed; we now use a first order filter.

Last paragraph on page 5

Similar as in the last comment also more details and justification on the symmetric term could be helpful for the reader. E.g. some more details and intuition on the role of the $\phi \phi^T$ homeostasis terms could be useful. Particularly as in earlier references to S1-1 this term is already used but in a non-symmetric way. It might be worth starting introducing these earlier so when jumping to the supplement things are already clear.

This has been changed, this appendix removed. We now focus the discussion on homeostasis, with a much broader view in the beginning of the main text.

Figure 3.

Axes have no scales, but this Figure is referred to later on page 16 asking the reader to compare scales !

The reference to this figure has been removed, but the axes were added to the figure.

Figure 4.

A,C, After learning there is a ringing in the Im part of the relevant eigenvalue (also visible in other figures). Where is this ringing coming from ?

This phenomenon doesn't exist with the first order filter.

The ringing of the eigenvalue was a result of the oscillatory dynamics of firing rates.

The caption can be more precise, particularly how the overlaps in B are calculated should be more explicitly stated.

These technical details are now explained in the Methods section.

D/ Why does the suppression of the memory saturate between 0.2 and 0.3 ?

This happens only with the second order filter. With the first-order filter the real part of the memory eigenvalue returns to the "bulk".

Section 2.2. Retention

In my view, how the attraction of the neuronal activity to the memory plane is created could be explained in more detail.

This section has been removed in the revised manuscript. We devote a separate section now to explain the implications of imaginary-coded memory, and there we discuss the dynamical properties of attractors in our model.

Page 11

As already stated above, the homeostasis term $f(x)g(x)^T$ should be better motivated, how could this term arise biologically?

We now motivate this mechanism as a firing-rate decorrelation. The revised paper contains extended discussion of the various homeostasis mechanisms used and their motivation.

Figure 5.

Is there a way to understand how large the asymmetry in the homeostasis term has to be in order to ensure stability?

Fig. 5 is removed in the new version. It turns out that the extent of asymmetry is not the crucial property for determining memory retention. This is now explained in detail in the manuscript and illustrated by applying an additional homeostasis mechanism with no special symmetry properties.

After eq. (5).

Sufficiently large ρ should be made more specific. Also ρ is related to the imprinting mechanism as stated on page 9.

It would be good to understand the relation between stimulus duration and amplitude and ρ more precisely, as this seems to determine whether the memory becomes a limit cycle.

These statements were indeed imprecise in the previous version. A closer inspection has revealed that the stability of the memory-related limit-cycle depends only on the real part of connectivity. In the revised manuscript we detail the criterion for stability in Methods and in a dedicated Supplementary section.

We also comment on the relationship between ρ and the memory properties in the main text:

“The strength of the memory representation - corresponding to the magnitude of the imaginary eigenvalue - depends monotonically on stimulation duration and on input amplitude.”

Figure 6.)

The figure and its caption on its own is confusing as different conditions lead to the different panels. Those should be stated clearly in the caption. The magnitudes in the various panels of the memory patterns differ, are of the order of $O(100)$ in A,B and $O(2)$ in C.

Caption A. Is a square root missing over the N in the coordinates p_u, p_v ?

Panel B. What happens if the overlap initially is much smaller? Further, what is the mechanisms that stabilizes certain memory components apart from the main one while others are continuing to decay ?

Panel C is confusing together with its caption in my view, it might be easier to explicitly show time on one axis by reducing the p coordinates to a radius as in B, or use a better time coding and reproduce that in D ?

Following these remarks, this figure has been heavily revised, and the points mentioned above were corrected.

Hoe do these simulation in Fig. compare to the reduced dynamics / theoretical predictions from the SI?

We have added these tests to the manuscript in Supplementary section S1. We show that the radius of the limit cycle is well approximated by the low-dimensional model over a broad range of parameters.

Page 15. Memory load: 2 dimensions are used to encode memory which would make one expect to find an memory bound alpha that is half the Hopfield bound ? Why is it still M/N ?

This is now better explained in the text, in Supplementary section S2.

Page 15. Ref to Fig 3B is either wrong or not appropriate as Fig 3 does not have any scales to compare to.

Reference removed.

Page 15, last paragraph before Discussion

The results here are quite interesting but would need a more thorough explanation in my view, same for SI-6.

Since the revised paper considers a broader view of more than one homeostasis mechanism, this point is less relevant and has been removed.

Discussion

The produced memory traces show oscillatory activity in a single pair of neurons in this model while neuronal oscillations typically require a much larger set of neurons. In my understanding the population activity of the model network studied here is not coherent on the population level and thus would not show any macroscopic oscillations. Further the oscillation frequency is related to the memory strength in my view in this model ? In my view the relation to the experimental studies is therefore too speculative. Do the experimentally observed oscillations really reflect the dynamics observed in this model ?

The oscillations in our model are a network phenomenon, and involve all neurons. This is similar to the case of a Hopfield model, in which all neurons participate in all memories. We thus expect to observe macroscopic oscillations, and indeed the new supplementary figure shows that such oscillations can be observed in our model. We thus think this is consistent with observations of

increased oscillatory activity during the recall of items from long-term to working memory. Furthermore, as the reviewer suggests, our model predicts a correlation between the oscillation frequency and memory strength – which is now highlighted in the supplementary figure. It turns out that this correlation is weaker than the between-memory variability in our model, and we discuss this and other possible predictions in the text.

Maybe the results on the 'limit cycle memories' generalize to larger dimensional stimuli and larger dimensional memory spaces, but would they still for simple limit cycles there or produce a much more complex dynamical state ?

The current model only supports two-dimensional limit cycles (embedded in the high dimensional space of all neurons). High-dimensional stimuli will be decomposed into separate limit cycles.

As pointed out in the intro to this report, the idea of orthogonality is not necessarily novel and in my view needs more discussion.

This is now addressed in the Discussion.

Making a connection to an experimental prediction is quite interesting and it might be beneficial to the manuscript to move Figure S6 to the main text.

We have further developed the discussion of experimental predictions, which appear in the Discussion section of the main text. However, we feel that the (new) figure and related details are better positioned in the Supplementary.

SI-S1

- $R^{N \times N}$ is not isomorphic to $R^{N^2 \times 1}$ but R^{N^2} only !

- S1 is referenced in the main text to before explaining much of the notation used here. It might be good to either explain it in S1 or reference to S1 later in the main manuscript.

This appendix has been removed.

SI-S2

The video needs more explanation, it is unclear what network is precisely simulated etc.

This appendix has been removed.

SI-Figure S3.

A few more data points on how the resonance frequency mismatch affects learning could be interesting, See also remarks to the results section above.

This appendix has been removed.

SI-S4

Derivation seems correct, but W is always wrongly defined as zero in the SI while I think $uv^T - vu^T$ is meant.

Fixed, thank you.

SI-S5

The link to the 4-cycle result is nice.

Thanks.

Figure S5: It is not clear to me how the figure is generated. Are the results averages of many networks and initial states ?

Where are the errors and how big are they on the curves ?

Why is the L1 radius used and not the L2 radius for m ? What is the most comparable overlap measure between the classical Hopfield network and the one used here?

Why is there so much more variance in the anti-symmetric model in the curves in general ?

As pointed out above, why is the limiting capacity at $\alpha = 1/2$ and not some smaller value as always two dimensions are used to store the memory ?

We now provide more detailed explanations on the computation, as well as more averaging.

Figure S5-2

It seems the anti-symmetric part in A and B performs worse (and the curves are much more fluctuating in comparison to the symmetric Hopfield network. The text states otherwise ?

Again no error bars and info on how the curves are obtained is given.

This figure has been modified and augmented with more statistics. Explanation on the methods has been provided.

SI-S6

Given the definition of $q \sim 1/\sqrt{N}$ in SI-S4, there is a factor of N missing in my opinion in the dynamics for δ and γ in front of the q and in the following equations ?

This section has been removed. (The factor was indeed missing).

SI-S7

The model predictions are interesting. Is there any way to get some analytic handle on those results ?

Predictions are now discussed at length, however we were not able to derive them analytically.

References

Several references seem to be wrongly cited or missing the last author.

Corrected.

References

Miner, D., Hoffmann, F. Z., Kleberg, F., & Triesch, J. (2017). Structural Plasticity and the Generation of Bidirectional Connectivity. In *The Rewiring Brain* (pp. 247-260).

Clopath, C., Büsing, L., Vasilaki, E., & Gerstner, W. (2010). Connectivity reflects coding: a model of voltage-based STDP with homeostasis. *Nature neuroscience*, 13(3), 344.

Reviewers' Comments:

Reviewer #1:

Remarks to the Author:

I would like to start by congratulating the authors for a substantially improved manuscript, and for addressing the scientific concerns that I had in the previous round of the review. Having said that, despite of the many improvements, the paper is not sufficiently clearly-written for the general audience of Nature Communications, most of them are not computational neuroscientists. More effort should be made to clarify the manuscript.

Abstract: "We find a simple and general principle stemming from stability arguments, that links eigenvalues in the complex plane to memories. Specifically, imaginary-coded memories are more resilient to noise and homeostatic plasticity than their real-coded counterparts." Without reading the paper first, these sentences are meaningless and confusing.

Line 14: "invariant features" is a term that may make sense to physicists but not to the general audience of the journal. The term invariance is used throughout the manuscript.

Figure 2: Despite a substantial effort, I did not understand, in details, the embedding of the memory in the matrix. This should be clarified.

Additional points:

Line 64-66: Motivating the "restraining mechanism" as a way of preventing divergence of the firing rate does not make sense when the firing rate non-linearity is tanh, which by itself prevents rate divergence.

I am surprised that the issue of sparse connectivity, which was raised by both reviewer 3 and I, was addressed in the rebuttal letter, but was left out from the manuscript. I strongly recommend addressing it, at least in the Supplementary Information section.

Reviewer #2:

Remarks to the Author:

The manuscript has been considerably improved. In particular, the authors now simulate a more realistic homeostasis process (whose goal is to maintain firing rate at some set value), though most of the figures still use a completely unrealistic process, given our current knowledge of synaptic dynamics.

However, there are still a number of issues that need to be discussed.

- The authors discuss more clearly why symmetric and antisymmetric perturbations on the connectivity matrix lead to different outcomes. In this discussion, the authors show that the decay of 'real-coded' memories will happen if the associated eigenvalue crosses the imaginary axis, therefore triggering synaptic dynamics that lead to the 'erasure' of the corresponding perturbation. However they don't discuss what happens if the perturbation does not destabilize the background state. What about a scenario in which a background state remains stable in spite of symmetric perturbations being added to the connectivity matrix? Such models could nonetheless exhibit stable attractor states correlated with the memories because of the non-linearity of the transfer function. This scenario would be implemented if in the background state neurons are at the 'foot' of their transfer function so the slope is low (preventing

destabilization). This is not the case in the current model, because in the 'background state' neurons are at the point where the transfer function has the highest slope. However, this is a particularly unrealistic feature of the model studied by the authors - in a background state one would expect neurons to fire at low rates and therefore to be at the foot of their transfer function. The paper thus leaves open the question of what would happen in a more realistic scenario, and in particular whether this strong difference between symmetric and antisymmetric perturbations would still exist.

- Sompolinsky (1986) studied the case of static noise added to a Hopfield connectivity matrix and showed it has a relatively weak effect on storage capacity - in fact when the noise is of the same order as the 'learned' part of the connectivity matrix, the capacity only decreases by about 50%. This suggests that a model with a stochastic ΔF (for instance, an Ornstein-Uhlenbeck process), and a symmetric ΔL with the same variance as ΔF would be consistent with data, and perform relatively well. This paper should be discussed, and the authors should explain this apparent discrepancy.

- In the Supplementary Information the authors claim that in the symmetric case the capacity has been computed only for binary neurons. They seem to have missed several papers on the capacity of associated memories with graded neurons, see Shiino and Fukai 1990 (J.Phys.A 1990 23:L1009-L1017) Kuhn et al 1991 (Phys Rev A 43:2084-2087), Amit and Tsodyks 1991 (Network 2: 275-294).

Minor issues:

- 'Such a structure adds a real eigenvalue to the spectrum' - technically the statement is incorrect since the number of eigenvalues does not change.

- Figure 2: In the caption, penultimate line (B) should be (D) and (D) should be (E).

- In Figure 5: Does the orange zoomed curve refer to t_2 or t_3 ?

- Supplementary Information, Eq. following Eq.(2): the second term in the argument of the sign function should be $p_v v_i$ not $p_u v_i$.

Reviewer #3:

Remarks to the Author:

The authors have substantially revised the manuscript.

By removing questionable parts of the model (second order filter) and adding a second homeostasis rule the manuscript has improved substantially. The authors have also addressed most of the earlier comments (see below).

The only two major points for criticism left are robustness of the results including relevance to biology

as well as precision.

Regarding the latter, all the assumptions made in the model could be pointed out more thoroughly in each step when introducing the model and the short-comings as well as validity could be discussed in the Discussion section.

Regarding robustness, I have a few concerns left (see comments below). If thoroughly addressed, the manuscript should be considered for publication.

Comments:

Robustness:

Figure 1 in the reply to the reviewers shows one example of how sparsity does not affect the results. While this might be the case in this example, I doubt that sparsity has not effect at all.

In particular, it is not clear how the sparsity is implemented (which is one example of the imprecision in the manuscript pointed out earlier). Are all directed connections chosen randomly with a probability p so that the resulting topological constraints are asymmetric (i.e. if a connection $i \rightarrow j$ is chosen to exist does this imply the existence of a connection $j \rightarrow i$?). If directed links are chosen independently in both directions I expect the mechanisms to be affected more than what the authors reply suggest. In my view, this is a critical point regarding robustness and biological relevance and should be discussed.

In the reply to the reviewers, a single example was given how sparsity affects robustness, but no statistics supporting this statement etc. are provided. I at least expect a minimal threshold for the sparsity below which the mechanism discussed in the manuscript is not sustainable. Where is it? How does it compare to connectivity statistics in biological networks?

In addition, how robust are the results when adding the additional constraint of Dale's principle where neurons can only be either excitatory or inhibitory? (i.e. weights can assume either only positive or only negative values but not both).

How does this biological constraint affect the dynamics and the results for memory encoding? Does the model result in certain

observable structures in excitatory vs inhibitory neurons?

I.e. is there more to predict about the network connectivity

from this model beyond the correlations of the weights in reciprocally connected neurons?

Clarity / Precision:

1.) Both homeostasis rules, while taken from other studies could be introduced in more detail for clarity.

2.)

Figure 2: From the caption (and even the text) it should be made more clear what exactly happens at the 'stimulation' or weight change at around 1/4 of the time traces in C-E.

While the authors use the term 'homeostasis' in the manuscript they use 'control' in the figure 2 B.

While these are not necessarily the same a link could be made here for easier access of the main idea.

Any axis should have a time scale in my opinion unless it is a sketch. Panels C-D are simulations and thus for reproducibility the time scale should be given (regardless of whether it's relevant or not).

3.) Line 130: I still believe the authors should not only state the result for their anti-symmetric learning rule derived in the SI but could give a bit more details how the smoothing γ arises in their formula (as described in the SI)

4.) SI: the authors show that specific correlations arise between reciprocal connections. It is left unclear what kind of effects one would see in standard learning rules that do not encode memories in the imaginary parts? I would still expect changes in the correlation structure of the weights between retrieval and learning. The differences should be made more clear in order to make a stronger case for experimental tests.

Dear editor and reviewers,

Thank you for your positive feedback and remarks concerning our revision.

We have addressed the remaining comments, as elaborated below, and believe that the manuscript now answers all concerns.

Sincerely

Lee, Naama and Omri.

Reviewer #1 (Remarks to the Author):

I would like to start by congratulating the authors for a substantially improved manuscript, and for addressing the scientific concerns that I had in the previous round of the review. Having said that, despite of the many improvements, the paper is not sufficiently clearly-written for the general audience of Nature Communications, most of them are not computational neuroscientists. More effort should be made to clarify the manuscript.

Abstract: "We find a simple and general principle stemming from stability arguments, that links eigenvalues in the complex plane to memories. Specifically, imaginary-coded memories are more resilient to noise and homeostatic plasticity than their real-coded counterparts." Without reading the paper first, these sentences are meaningless and confusing.

These sentences have been replaced with a less obscure phrasing of the results. More generally we have made every effort to clarify the writing for a general audience.

Line 14: "invariant features" is a term that may make sense to physicists but not to the general audience of the journal. The term invariance is used throughout the manuscript.

The concept of invariant features at the network level is a central one in the paper. We have used the analogous phrasing "stability" to expose this concept and define it more precisely in the introduction.

Figure 2: Despite a substantial effort, I did not understand, in details, the embedding of the memory in the matrix. This should be clarified.

This is now better explained both in the text and in the figure caption.

Additional points:

Line 64-66: Motivating the "restraining mechanism" as a way of preventing divergence of the firing rate does not make sense when the firing rate non-linearity is tanh, which by itself prevents rate divergence.

Corrected. A restraining mechanism is needed to prevent divergence of synaptic weights. In addition, homeostasis has to ensure stability of firing rates.

It is true that saturating non-linearity would prevent divergence of firing rates even in an unstable regime, but it is a reasonable and common assumption that neurons are further

restrained not to reach this bound (e.g., neurons fire at rates much lower than dictated by refractory periods).

I am surprised that the issue of sparse connectivity, which was raised by both reviewer 3 and I, was addressed in the rebuttal letter, but was left out from the manuscript. I strongly recommend addressing it, at least in the Supplementary Information section.

A supplementary section was added with these results, which are also mentioned in the Results section of the revised manuscript.

Reviewer #2 (Remarks to the Author):

The manuscript has been considerably improved. In particular, the authors now simulate a more realistic homeostasis process (whose goal is to maintain firing rate at some set value), though most of the figures still use a completely unrealistic process, given our current knowledge of synaptic dynamics.

Actually Fig. 5, which contains the integration of all phases and demonstrates stable learning with unstable synapses, shows results of simulation with the more realistic mechanism. This is now more clearly described in the text.

However, there are still a number of issues that need to be discussed.

- The authors discuss more clearly why symmetric and antisymmetric perturbations on the connectivity matrix lead to different outcomes. In this discussion, the authors show that the decay of 'real-coded' memories will happen if the associated eigenvalue crosses the imaginary axis, therefore triggering synaptic dynamics that lead to the 'erasure' of the corresponding perturbation. However they don't discuss what happens if the perturbation does not destabilize the background state. What about a scenario in which a background state remains stable in spite of symmetric perturbations being added to the connectivity matrix? Such models could nonetheless exhibit stable attractor states correlated with the memories because of the non-linearity of the transfer function. This scenario would be implemented if in the background state neurons are at the 'foot' of their transfer function so the slope is low (preventing destabilization). This is not the case in the current model, because in the 'background state' neurons are at the point where the transfer function has the highest slope. However, this is a particularly unrealistic feature of the model studied by the authors - in a background state one would expect neurons to fire at low rates and therefore to be at the foot of their transfer function. The paper thus leaves open the question of what would happen in a more realistic scenario, and in particular whether this strong difference between symmetric and antisymmetric perturbations would still exist.

We assume that the background state arises from a balance between exciting and controlling forces. Specifically, we assume the existence of some homeostatic mechanism that stabilizes firing activity and that it is necessary to do so. This is a standard assumption in light of the rich and complex nature of spontaneous activity. We do not assume that the instability of the background state, which requires homeostasis, is a result of storing information in the symmetric component. In fact, the same instability is there when we store information in the imaginary component as well.

It is possible to conceive a situation in which the background state is completely quiescent. In such a case, there is no need of a homeostatic mechanism, and therefore we do not expect any difference between the resilience of real- and imaginary- coded memories.

- Sompolinsky (1986) studied the case of static noise added to a Hopfield connectivity matrix and showed it has a relatively weak effect on storage capacity - in fact when the noise is of the same order as the 'learned' part of the connectivity matrix, the capacity only decreases by about 50%. This suggests that a model with a stochastic ΔF (for instance, an Ornstein-Uhlenbeck process), and a symmetric ΔL with the same variance as ΔF would be consistent with data, and perform relatively well. This paper should be discussed, and the authors should explain this apparent discrepancy.

The apparent discrepancy is because the noise in [Sompolinsky, 1986] was static, whereas we consider a dynamic noise source. This is why the dissipative term in the Ornstein-Uhlenbeck process erases the memory item, as shown in Fig. 2A. Without the dissipative term, synaptic weights would diverge due to the noise process. This point is now highlighted in the Discussion section of the revised manuscript, where this paper is also discussed.

- In the Supplementary Information the authors claim that in the symmetric case the capacity has been computed only for binary neurons. They seem to have missed several papers on the capacity of associated memories with graded neurons, see Shiino and Fukai 1990 (J.Phys.A 1990 23:L1009-L1017) Kuhn et al 1991 (Phys Rev A 43:2084-2087), Amit and Tsodyks 1991 (Network 2: 275-294).

Thank you for pointing this out. We now mention this literature, and also numerically evaluate the critical capacity of our model in the continuous case.

Minor issues:

- 'Such a structure adds a real eigenvalue to the spectrum' - technically the statement is incorrect since the number of eigenvalues does not change.

Fixed

- Figure 2: In the caption, penultimate line (B) should be (D) and (D) should be (E).

Fixed

- In Figure 5: Does the orange zoomed curve refer to t_2 or t_3 ?

t_2 , thank you. Fixed

- Supplementary Information, Eq. following Eq.(2): the second term in the argument of the sign function should be $p_v v_i$ not $p_u v_i$.

Fixed

Reviewer #3 (Remarks to the Author):

The authors have substantially revised the manuscript.

By removing questionable parts of the model (second order filter) and adding a second homeostasis rule the manuscript has improved substantially. The authors have also addressed most of the earlier comments (see below).

The only two major points for criticism left are robustness of the results including relevance to biology as well as precision.

Regarding the latter, all the assumptions made in the model could be pointed out more thoroughly in each step when introducing the model and the short-comings as well as validity could be discussed in the Discussion section.

Regarding robustness, I have a few concerns left (see comments below). If thoroughly addressed, the manuscript should be considered for publication.

Comments:

Robustness:

Figure 1 in the reply to the reviewers shows one example of how sparsity does not affect the results. While this might be the case in this example, I doubt that sparsity has not effect at all.

In particular, it is not clear how the sparsity is implemented (which is one example of the imprecision in the manuscript pointed out earlier). Are all directed connections chosen randomly with a probability p so that the resulting topological constraints are asymmetric (i.e. if a connection $i \rightarrow j$ is chosen to exist does this imply the existence of a connection $j \rightarrow i$?). If directed links are chosen independently in both directions I expect the mechanisms to be affected more than what the authors reply suggest. In my view, this is a critical point regarding robustness and biological relevance and should be discussed.

In the reply to the reviewers, a single example was given how sparsity affects robustness, but no statistics supporting this statement etc. are provided. I at least expect a minimal threshold for the sparsity below which the mechanism discussed in the manuscript is not sustainable. Where is it?

How does it compare to connectivity statistics in biological networks ?

We have added an Appendix addressing these points, and in particular added more statistics to the calculations, showing the limits of validity in terms of sparseness.

In addition, how robust are the results when adding the additional constraint of Dale's principle where neurons can only be either excitatory or inhibitory? (i.e. weights can assume either only positive or only negative values but not both).

How does this biological constraint affect the dynamics and the results for memory encoding ? Does the model result in certain observable structures in excitatory vs inhibitory neurons ? I.e. is there more to predict about the network connectivity from this model beyond the correlations of the weights in reciprocally connected neurons?

In our model excitatory and inhibitory connections exist because of the random nature of the matrices used. However, adding the precise structure of Dale's law requires defining inhibitory

and excitatory neuronal populations and adds many new parameters to the model. This realistic feature (as well as other features, such as spiking neurons) remain outside the scope of the current work and await future investigation.

Clarity / Precision:

1.) Both homeostasis rules, while taken from other studies could be introduced in more detail for clarity.

We have made every effort to clarify the introduction of these rules in the revised manuscript.

2.)

Figure 2: From the caption (and even the text) it should be made more clear what exactly happens at the 'stimulation' or weight change at around 1/4 of the time traces in C-E.

This is now better explained both in the text and in the figure caption.

While the authors use the term 'homeostasis' in the manuscript they use 'control' in the figure 2 B.

While these are not necessarily the same a link could be made here for easier access of the main idea.

This is explained circa line 74.

Any axis should have a time scale in my opinion unless it is a sketch. Panels C-D are simulations and thus for reproducibility the time scale should be given (regardless of whether it's relevant or not).

Fixed.

3.) Line 130: I still believe the authors should not only state the result for their anti-symmetric learning rule derived in the SI but could give a bit more details how the smoothing γ arises in their formula (as described in the SI)

Additional information has been added to the description of the learning rule in the main text, and to the derivation in the Methods section.

4.) SI: the authors show that specific correlations arise between reciprocal connections. It is left unclear what kind of effects one would see in standard learning rules that do not encode memories in the imaginary parts? I would still expect changes in the correlation structure of the weights between retrieval and learning. The differences should be made more clear in order to make a stronger case for experimental tests.

Indeed, even within our framework the exact effect on correlations depends on the specifics of the homeostasis mechanism. We agree with the reviewer that a symmetric learning rule might also result in some change of correlations. We therefore agree that, contrary to our initial intuition, measuring correlations is of little predictive value. In contrast, the existence of an

oscillatory component is more indicative of our class of models. This is now stated more clearly in the Discussion section of the revised manuscript.

Reviewers' Comments:

Reviewer #2:

Remarks to the Author:

The paper has been further improved and I believe it can now be published in present form.

Reviewer #3:

Remarks to the Author:

In the revised version of the manuscript the authors are more careful with their statements and the manuscript is more clear now.

On the positive side, focusing on 'dynamic memories' is interesting and deserves more attention in the community.

On the negative side, while the theory is interesting, I honestly don't think the manuscript makes valuable experimental predictions in its current form for the following reasons:

- 1.) As the authors point out by themselves and even counter to their own intuition, the correlation structure changes are not unique to their model and can arise in so many ways it will be hard to nail down the precise mechanisms.
- 2.) Oscillatory activity arises generically in models with negative feedback and thus cannot serve as a good indicator of the authors memory mechanisms either. Even the correlation of memory strength and oscillatory frequency will be very hard to discriminate, given the multi-component oscillatory activity in the brain generated by various often already well described mechanisms.
- 3.) Measuring the effect on the network level is not a feasible prediction either as to date it is not even clear how memories distributed across brain areas. On the long run this might be resolved but the authors claim in the abstract and manuscript of making experimental measurable predictions is not correct in my opinion and should be removed.

Of course discussing possible routes to measure the effect is valuable and should be discussed.

Point-by-point response.

REVIEWERS' COMMENTS:

Reviewer #2 (Remarks to the Author):

The paper has been further improved and I believe it can now be published in present form.

Reviewer #3 (Remarks to the Author):

In the revised version of the manuscript the authors are more careful with their statements and the manuscript is more clear now.

On the positive side, focusing on 'dynamic memories' is interesting and deserves more attention in the community.

On the negative side, while the theory is interesting, I honestly don't think the manuscript makes valuable experimental predictions in its current form for the following reasons:

1.) As the authors point out by themselves and even counter to their own intuition, the correlation structure changes are not unique to their model and can arise in so many ways it will be hard to nail down the precise mechanisms.

Indeed correlation structure changes are not unique, and this is not presented as a prediction.

2.) Oscillatory activity arises generically in models with negative feedback and thus cannot serve as a good indicator of the authors memory mechanisms either. Even the correlation of memory strength and oscillatory frequency will be very hard to discriminate, given the multi-component oscillatory activity in the brain generated by various often already well described mechanisms.

Thanks to this remark we have elaborated in Supplementary Note 5 on the ability to discriminate the relation between memory strength and frequency. In fact, while oscillations do appear in other contexts, this relation is a clear signature of our model and of the fact that the memory itself is dynamic. The new figure now shows that by examining the normalized power of the memory item it is feasible to discriminate this relation.

3.) Measuring the effect on the network level is not a feasible prediction either as to date it is not even clear how memories distributed across brain areas. On the long run this might be resolved but the authors claim in the abstract and manuscript of making experimental measurable predictions is not correct in my opinion and should be removed.

Of course discussing possible routes to measure the effect is valuable and should be discussed.

Formulation of this claim has been modified – we agree that it is not a prediction.